

# Contrail formation on ambient aerosol particles for aircraft with hydrogen combustion: A box model trajectory study

Andreas Bier[1], Simon Unterstrasser[1], Josef Zink[1], Dennis Hillenbrand[1], Tina Jurkat-Witschas[1], and Annemarie Lottermoser[1]

[1]Deutsches Zentrum für Luft- und Raumfahrt, Insitut für Physik der Atmosphäre, Oberpfaffenhofen, Germany

**Correspondence:** Andreas Bier (Andreas.Bier dlr.de)

**Abstract.**

Future air traffic using (green) hydrogen ($H_2$) promises zero carbon emissions but the effects of contrails from this new technology has hardly been investigated. We study contrail formation behind aircraft with $H_2$ combustion by means of the particle-based Lagrangian Cloud Module (LCM) box model. Assuming the absence of soot and ultrafine volatile particle for-
mation, contrail ice crystals form solely on atmospheric background particles mixed into the plume. While a recent study extended the original LCM with regard to the contrail formation on soot particles, we further advance the LCM to cover the contrail formation on ambient particles. For each simulation, we perform an ensemble of box model runs using the dilution along 1000 different plume trajectories that are based on 3D Large Eddy Simulations using the FLUDILES solver.

The formation threshold temperature of $H_2$ contrails is by around $10\,\mathrm{K}$ higher than for conventional contrails (which form be-
hind aircraft with kerosene combustion) due to a factor of 2.6 higher energy-specific water vapor emission. Therefore, contrail formation becomes primarily limited by the homogeneous freezing temperature of the water droplets formed on the ambient particles such that contrails can form at temperatures down to around $234\,\mathrm{K}$.

The number of formed ice crystals varies strongly with ambient temperature even far away from the contrail formation thresh-old. The latter is because the water-supersaturation in the plume lasts longer for colder conditions and, hence, more of the
entrained aerosol particles can form droplets and ice crystals. The contrail ice crystal number clearly increases for a higher ambient aerosol number concentration. The increase becomes weaker for higher number concentrations ($>\approx 200\,\mathrm{cm}^{-3}$) and lower ambient temperatures ($< 230\,\mathrm{K}$). The ice crystal number decreases significantly for ambient particles with mean dry radii $<\approx 10\,\mathrm{nm}$ due to the Kelvin effect.

Besides simulations with one aerosol particle ensemble, we analyze contrail formation scenarios with two co-existing aerosol
particle ensembles that differ either in their mean dry size or hygroscopicity parameter. We compare them to scenarios with a single ensemble that is the average of the two ensembles. We find that the total ice crystal number can differ significantly between the two cases, in particular if nucleation mode particles are involved.

Due to the absence of soot particle emissions, the ice crystal number in $H_2$ contrails is typically reduced by more than 80–90% compared to conventional contrails. The contrail optical thickness is significantly reduced and $H_2$ contrails either become later
visible than kerosene contrails or are not visible at all for low ambient particle number concentrations. On the other hand, $H_2$ contrails can form at lower flight altitudes where conventional contrails would not form.



## 1   Introduction

The contribution of aviation to the total anthropogenic climate forcing is estimated to be around 3.5% (Lee et al., 2021). Besides the aircraft $CO_2$ emissions, contrail cirrus have a large contribution to the aviation radiative forcing (e.g., Boucher et al.,

2013; Bock and Burkhardt, 2016b; Bier and Burkhardt, 2022). There are several measures to mitigate the climate impact due to contrail cirrus. One mitigation option is reducing the number of formed contrail ice crystals, which strongly impact the further contrail cirrus life cycle and the radiative forcing (e.g., Unterstrasser and Gierens, 2010; Bier et al., 2017; Burkhardt et al., 2018). This might be achieved by reducing soot particle number emissions, since contrail ice crystals form in particular on soot particles for conventional passenger aircraft engines (e.g., Kärcher and Yu, 2009; Kleine et al., 2018). Several ground and flight

measurement campaigns have shown significant reductions in engine soot number emissions by using alternative fuel blends with a lower aromatic content (e.g., Moore et al., 2017; Voigt et al., 2021; Bräuer et al., 2021). Switching from the reference Jet A-1 fuel to semisynthetic or biofuel blends, Voigt et al. (2021) and Bräuer et al. (2021) also find significant reductions in young contrail ice crystal numbers by around 20–70%. Burkhardt et al. (2018) and Bier and Burkhardt (2022) emphasize that there is a strong non-linearity between the global contrail cirrus radiative forcing and the young contrail ice crystal number.

Hence, even larger reductions in the number of formed ice crystals are desirable to obtain a substantial mitigation effect.
(Green) hydrogen ($H_2$) combustion is a promising technology to reduce the overall aviation climate impact. It provides around 3 times more energy per fuel mass than kerosene fuel (Najjar, 2013), but it delivers much less energy by volume at typical atmospheric conditions. Hence, $H_2$ is typically brought to liquid phase at $20\,K$ and stored in special tanks of the cryoplane. During $H_2$ combustion, the main emission product is water vapor and its emission is roughly a factor of 2.6 larger compared

to kerosene for the same amount of released combustion energy and a similar propulsion efficiency (e.g., Schumann, 1996). Increased water vapor emissions in the stratosphere would cause a significant radiative warming (Pletzer et al., 2022), but this impact would be low as long as the aircraft fly at altitudes in the troposphere (e.g., Wilcox et al., 2012). While $NO_x$ is still produced due to high flame temperatures, we expect neither direct $CO_2$ nor soot particle emissions during $H_2$ combustion. However, it was observed in laboratory studies that the emission of lubricant oil vapors can lead to the formation of ultrafine

volatile particles (Ungeheuer et al., 2022). Up to now, measurements on $H_2$ contrails do not exist. Airbus and DLR are planning measurements behind a glider equipped with a small $H_2$ combustion engine within the Blue Condor campaign (Airbus, 2022). Moreover, Airbus aims at establishing the world's first commercial aircraft based on hydrogen propulsion by 2035 within the "ZEROe" project. Marquart et al. (2005) and Ponater et al. (2006) estimated the radiative forcing (RF) of line-shaped contrails for a hypothetical fleet of cryoplanes in comparison with a conventional fleet within a global climate model (GCM). They

found similar RF values for both types of fleets. The decrease in optical thickness for $H_2$ contrails (RF down) was roughly balanced by the larger contrail coverage (RF up). This estimate is based on a simple parameterization of line-shaped contrails (Ponater et al., 2002), where e. g. the contrail cover scales with the contrail formation frequency and the ice water content is simply diagnosed by the atmospheric water vapor available for deposition. Recent GCM contrail parameterizations are more advanced as they simulate the full contrail (cirrus) life cycle, treat contrails as a separate cloud class to natural clouds and

introduce contrail ice water content, coverage and ice crystal number as prognostic variables (Bock and Burkhardt, 2016a; Bier





and Burkhardt, 2022).

The hot exhaust plume behind the aircraft engines continuously expands and cools due to entrainment of ambient air. Under certain atmospheric conditions and depending on specific engine/fuel parameters, the plume humidity temporally surpasses water-saturation in the early jet phase and enables the formation of contrails. This condition is described by the Schmidt-

Appleman (SA)-criterion (Schumann, 1996) which is based purely on the thermodynamics of the plume mixing process. If the SA-criterion is fulfilled, plume particles can activate into water droplets (e.g., Kärcher and Yu, 2009; Kärcher et al., 2015). They subsequently turn to ice crystals by homogeneous freezing if ambient temperature is below the homogeneous freezing temperature. Switching to $H_2$ combustion with expected soot-free emissions, contrail ice crystals can still form on upper tropospheric (UT) background particles that are entrained into the plume (e.g., Kärcher et al., 1996, 2015).

Some recent box model studies and analytical approaches (Kärcher and Yu, 2009; Kärcher et al., 2015; Bier and Burkhardt, 2019) have already included ice crystal formation on ambient particles mixed into the plume. They show that this process will become relevant if soot number emissions from conventional aircraft engines are reduced by at least 2 orders of magnitude (refered to as "soot-poor emissions"). Kärcher (2018) estimate a decrease in contrail ice crystal number by around 1-2 orders of magnitudes when switching from conventional to soot-poor emissions at ambient temperatures for which ice crystals cannot

form on ultrafine volatile particles. While those studies in general assumed fixed ambient particle properties, Lewellen (2020) investigated contrail formation on ambient aerosol (besides soot and volatile particles) in a box model and Large Eddy Simulations (LES) and varied the ambient aerosol number concentration. Finally, we expect a high uncertainty in the estimated $H_2$ contrail ice crystal number due to a large variability in atmospheric particle properties (e.g., Minikin et al., 2003; Hermann et al., 2003; Brock et al., 2021; Voigt et al., 2022), which has not been examined in sufficient detail before.

The contrail formation studies mentioned in the preceding paragraph have been performed only for fuel/engine parameters that represent the kerosene case and considered the competition between ambient aerosol, soot and volatile particles. Ström and Gierens (2002) is the only contrail evolution study considering aircraft with $H_2$ combustion. They performed 2D simulations of young contrails including the contrail formation process in the jet phase. They prescribe a bi-modal log-normal aerosol size distribution and vary the aerosol number concentration as the most relevant input parameter. They employ a bulk approach for

the treatment of the ice microphysics and simulate the homogeneous freezing on wetted aerosol particles. They do not use any solubility model and simply assume that the background aerosol particles are composed of ammonium sulfate.

In the present study, we aim at providing a basic understanding of the processes regarding the contrail formation on ambient particles ("$H_2$ contrails"). Moreover, we will highlight main differences compared to conventional contrails where ice crystals mainly form on soot particles. We will also explain the impact of the increased water vapor emission due to $H_2$ combustion

on the contrail formation criterion and the thermodynamic plume properties. Our main objective is to explore the variability in contrail properties (in particular ice crystal number) due to the variability in atmospheric parameters on the one hand and due to the variability in ambient aerosol particle properties on the other hand. While previous studies focused only on the variation of the aerosol number concentration, we also investigate the impact of the mean aerosol dry size and the solubility. Moreover, we will analyze the impact of the competition of two co-existing ambient aerosol particle ensembles instead of a single one on





contrail ice nucleation. Finally, we will compare the number of formed ice crystals and optical thickness of $H_2$ contrails with conventional contrails.

## 2  Background and state of the art

This section provides a basic summary over the observed and modeled aerosol particle properties and then explains the impact of $H_2$ combustion on the thermodynamic contrail formation criterion.

### 2.1  Observed and modeled aerosol particle properties

The major source of UT aerosol particles are natural and anthropogenic emissions of gaseous aerosol precursors that are transported from lower altitudes by vertical updrafts like synoptic scale lifting or deep convection (e.g., Minikin et al., 2003), and form particles due to chemical ion nucleation (e.g., Lee et al., 2003). Another important source is the in-situ formation, caused by mixing processes and aircraft emissions (e.g., Hermann et al., 2003). Aviation contributes about 30–40% of the particle

number concentration in the northern mid-latitudes' UT between 7 and 12 km (Righi et al., 2013). The major relevance of ambient aerosol particles for contrails is likely over the high density air traffic regions like Central Europe, the Eastern USA and North Atlantic where contrails frequently form. This relevance will increase in the near future when first hydrogen engines become available (Righi et al., 2016). On the other hand, future atmospheric conditions are likely to have a reduced aerosol content due to the long term pursuit of a cleaner atmosphere (Andreae et al., 2005).

Currently, there are still few observations of UT aerosol particle properties available and we here provide a short summary of some important measurement campaigns: Minikin et al. (2003) investigated within two flight campaigns spatial distributions and vertical profiles of aerosol number concentrations both over the northern hemisphere (NH) and over the mid-latitudes of the southern hemisphere (SH) in the UT. As displayed in their Tab. 1, the measured number concentrations in the Aitken mode range from 130 to 400 $\mathrm{cm}^{-3}$ (290 to 9600 $\mathrm{cm}^{-3}$) in the SH (NH) those in the accumulation mode from 6 to 43 $\mathrm{cm}^{-3}$ (24

to 480 $\mathrm{cm}^{-3}$) in the NH (SH). In several measurement flights, Petzold et al. (2002) observed aerosol particle properties over eastern Germany in summer 1998 at altitudes from ground level to 11 km within the "Lindenberg Aerosol Characterization Experiment (LACE 98)". In addition to number concentrations, they derived aerosol particle size distributions at different altitudes (see their Fig. 5). In the considered UT and tropopause region, the smallest measured particle sizes (radius $\approx$ 50 nm) were the most abundant. Large data sets of aerosol particle number densities were acquired in the UT/lower stratosphere (LS)

in the subtropics, the tropical tropopause and the mid-latitudes during the "SCOUT-O3", "SCOUT-AMMA" and "TROCCI-NOX" campaigns (Borrmann et al., 2010). They reveal a large variability with number densities between 100 and more than 1000 $\mathrm{cm}^{-3}$ in the altitude range of 9 to 12 km. Brock et al. (2021) performed in-situ measurements of aerosol properties as part of the Atmospheric Tomography Mission (ATom) from 2016-2018 in particular over the Atlantic and Pacific Ocean. They show in their Fig. 12 vertical profiles of aerosol number concentration as well as fitted log-normal geometric diameter and

geometric width of the size distribution for the Nucleation, Aitken, Accumulation and Coarse mode particles. Additionally, Cloud Condensation Nucleii (CCN)-concentrations were measured at different water-supersaturations (Fig. 14), which show



a slight increase in the UT above 10 km and vary strongly with latitude. While these measurements are the most recent and comprehensive, they were mainly taken outside the main air traffic regions. (Beer et al., 2020) compared aerosol profiles above the North American continent and Europe to model data. They show (in their Supplement) altitude profiles of number con-
centrations with average values between 200 and 300 cm$^{-3}$ for non-volatile dry radii > 2.5 nm. Recently, long term aerosol measurements from a commercial aircraft platform within the "Civil Aircraft for Regular Investigation of the Atmospheric Based on an Instrument Container (CARIBIC)" project (Hermann et al., 2003) were compared with measurements over Europe during the Covid-19 pandemic (Voigt et al., 2022). Due to massive reductions in aviation and industrial emissions during the pandemic, significant reductions in aerosol number concentrations were observed in the UT potentially reflecting future
low emission scenarios.

The chemical composition of aerosol particles is of great importance and impacts several microphysical processes like hygroscopic growth and activation into water droplets. Liu et al. (2014) investigated hygroscopic properties of CCN based on their chemical composition in the North China Plain. They derived the hygroscopicity parameter ($\kappa$), introduced in the solubility model from Petters and Kreidenweis (2007), of 16 relevant inorganic salts and sulfuric acid. Thereby, a higher $\kappa$-value is asso-
ciated with a better solubility of the aerosol species. Sulfuric acid and most of the inorganic salts have $\kappa > 0.5$. In other studies, the $\kappa$-value of water soluble organic carbon is estimated to be around 0.3 (e.g., Padro et al., 2010) and that of freshly emitted aviation soot is close to zero (e.g Petzold et al., 2005; Kärcher et al., 2015). Pre-activated soot particles (e. g., by contrail ice in their pores) can be more water-soluble but also serve as heterogeneous ice nucleii (e.g., Marcolli, 2017). Composition measurements using single particle mass spectrometry investigating the size resolved mixing state of aeorosol have gained much
attention and provide the source for estimates on the hygroscopicity of background aerosol in the UT/LS (Froyd et al., 2019; Tomsche et al., 2022; Schmale et al., 2010).

Besides observation campaigns, climate models with aerosol physics (e.g., Stier et al., 2005; Kaiser et al., 2019) have been developed to simulate chemical formation and microphysical processes of aerosol particles. These models have been evaluated with observations and can be used for the investigation of the global aerosol climatology (e.g., Beer et al., 2020). Among
others, they highlight the large spatio-temporal variability of aerosol particle properties in terms of their number concentration, size distribution and chemical composition. In this work, we investigate the sensitivity to these parameters and their relevance for H$_2$ contrail properties.

## 2.2 Contrail formation criterion

Behind an aircraft engine, the hot and moist plume air mixes with the colder ambient air and the plume is continuously diluted. The so called "mixing line" describes the linear dependency between the partial vapor pressure and excess temperature in the plume. The Schmidt-Appleman (SA)-criterion is fulfilled for a sufficiently low ambient temperature such that the mixing line crosses the saturation vapor pressure over liquid water and hence the plume becomes water-supersaturated in a particular time period (Schumann, 1996). It is a necessary condition for contrail formation and has been empirically validated by several flight
campaigns for kerosene combustion (e.g., Busen and Schumann, 1995; Schumann et al., 2002). The SA-threshold temperature



($\Theta_\mathrm{G}$) is the largest ambient temperature for which water saturation is still reached in the plume (Schumann, 1996). It depends on the ambient relative humidity over water and the slope of the mixing line

$$G = \frac{EI_\mathrm{v}\, c_\mathrm{p}\, p_\mathrm{a}}{0.622\, Q(1-\eta)}, \tag{1}$$

where $c_\mathrm{p}$ is the specific heat capacity, $p_\mathrm{a}$ the ambient pressure, $EI_\mathrm{v}$ is the exhaust water vapor (mass) emission index, $Q$ the specific combustion heat, $\eta$ the propulsion efficiency and $EI_\mathrm{v}/Q$ is called the energy-specific water vapor emission index. The calculation of $\Theta_\mathrm{G}$ is described in Appendix of Schumann (1996).

| fuel/engine parameters | kerosene | hydrogen | ratio |
|---|---|---|---|
| $EI_\mathrm{v}$ / kg kg$^{-1}$ | 1.26 | **8.94** | 7.10 |
| $Q$ / MJ kg$^{-1}$ | 43 | **120** | 2.79 |
| $EI_\mathrm{v}/Q$ / kg MJ$^{-1}$ | 0.029 | **0.075** | 2.57 |
| $\eta$ | 0.36 | **0.36** | 1 |

**Table 1.** Fuel and engine parameters for kerosene (2nd column), hydrogen propulsion (3rd column) and the ratio between both (last column). The water vapor mass emission index and specific combustion heat are based on Tab. 1 of Schumann (1996). The propulsion efficiency is fixed for both fuel types to a value typical of an A340 aircraft according to Vancassel et al. (2014) and Bier et al. (2022).

Fig. 1 shows that the SA-threshold temperature generally increases with rising relative humidity over water ($RH_\mathrm{wat}$) on the one hand and with increasing ambient pressure on the other hand. The parts of the curves lying above the solid black line depict the ice-supersaturated cases supporting persistent contrails. In the following, we compare $\Theta_\mathrm{G}$ for hydrogen combustion (blue lines) with those for kerosene combustion (red lines). Using the parameters from Tab. 1, $EI_\mathrm{v}$ is around 7.1 and $Q$ 2.8 times higher for the hydrogen than for the kerosene case (see also Tab. 1). This leads to an overall increase in the slope of the mixing line (Eq. (1)) by a factor of $EI_\mathrm{v}/Q \approx 2.6$ for fixed $\eta$ and ambient pressure. As a consequence, $\Theta_\mathrm{G}$ is by around 10 K larger for the hydrogen than for the kerosene case (for otherwise fixed conditions). Considering the same atmospheric conditions and ensuring that $T_\mathrm{a} < \Theta_\mathrm{G}$, this will cause significantly higher (peak) plume water-supersaturation for the hydrogen case in the early jet phase because the difference $\Delta T = |T_\mathrm{a} - \Theta_\mathrm{G}|$ is accordingly higher (e.g., Kärcher et al., 2015; Bier et al., 2022). Moreover, droplet formation on aerosol particles will be enabled at higher ambient temperatures as we will show in the results section.

## 3 Methods

First, Sect. 3.1 gives an overview over the Lagrangian Cloud Module (LCM) and Sect. 3.2 describes the employed trajectory data and plume thermodynamics. Sect. 3.3 explains the basic contrail formation pathway on ambient particles and the associated





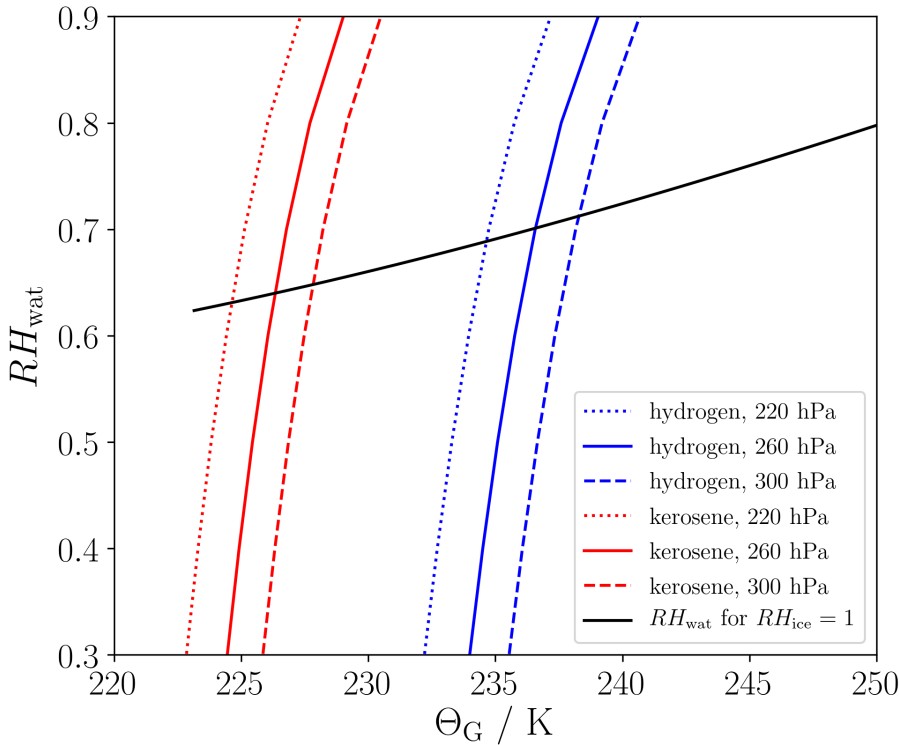

**Figure 1.** SA-threshold temperature ($\Theta_G$) versus relative humidity over water ($RH_{wat}$) for three ambient pressures (differentiated by the line style) and for the kerosene (red) and hydrogen (blue) engine parameters as they are defined in Tab. 1 each. The black solid line displays those $RH_{wat}$ that would result in ice saturation assuming ambient temperature equal to $\Theta_G$.

extension of the LCM based box model. Finally, Sect. 3.4 gives an overview over the box model settings and the baseline conditions for the $H_2$ combustion scenario.

### 3.1 LCM box model

LCM is a particle-based microphysical model that includes aerosol, droplet and ice microphysics (Sölch and Kärcher, 2010).
185 This particle-based approach has several numerical and physical advantages over common grid-based approaches, which are typically used in Computational Fluid Dynamics (CFD). It has been used for the simulation of natural cirrus clouds (e.g., Sölch and Kärcher, 2011), young contrails (e.g., Unterstrasser, 2014) and aged contrail-cirrus (e.g., Unterstrasser et al., 2017a). Recently, it has been extended by contrail formation microphysics on soot particles (Bier et al., 2022). Aerosol particles and hydrometeors are described by simulation particles (SIPs). Each SIP represents a certain number of aerosol particles/droplets/ice
190 crystals with the same properties and contains information about the liquid/ice water mass, radius, phase and particle type,





among others. These properties may change due to microphysical processes like hygroscopic growth of aerosol particles and activation into water droplets, condensational droplet growth, homogeneous freezing of super-cooled droplets, depositional ice crystal growth, latent heat release, aggregation of ice crystals, sedimentation and radiative effects. In this study, we will consider only those processes that are relevant for contrail formation and exclude aggregation, sedimentation and radiative effects.

## 3.2  Trajectory data and plume evolution

In a box model approach, fluid dynamics is not resolved and changes of thermodynamic properties inside the box are externally prescribed. We use the general plume dilution equations, which are described in Sect. 2.3.1 of Bier et al. (2022), to calculate the cooling and expansion of the plume as well as the evolution of the humidity. Based on 3D Large Eddy Simulations (LES) using the FLUDILES solver, Vancassel et al. (2014) sampled an aircraft plume with 25000 trajectories behind the engine of an A340-300 aircraft. Thereby, the temperature evolution $T_{\text{3D},k}(t)$ has been tracked for each trajectory indexed by $k$. As in Bier et al. (2022), we use these data to infer the plume dilution factor by assuming that temperature is a passive tracer

$$\mathcal{D}_k(t) = \frac{T_{\text{3D},k}(t) - T_{\text{3D,a}}}{T_{\text{3D},0} - T_{\text{3D,a}}}, \tag{2}$$

where $T_{\text{3D},0} = 580\,\text{K}$ and $T_{\text{3D,a}} = 220\,\text{K}$ are the plume exit and ambient temperature of the FLUDILES simulation.

In the following, we describe some modifications of the FLUDILES trajectory data set compared to the original one from Vancassel et al. (2014):

- As in Bier et al. (2022), we introduce a lower limit $T_{\text{3D},k}(t) + \epsilon$ and all $T_{\text{3D},k}(t)$-values below this lower limit are set to this value. We choose $\epsilon = 0.2\,\text{K}$, such that the implied dilution and plume area are consistent with the area enclosed by the trajectories.

- We have smoothed the time evolution of $T_{\text{3D},k}$ for each trajectory such that $T_{\text{3D},k}$ becomes a monotonically decreasing function with increasing plume age. This means we set $T_{\text{3D},k}(t) := \text{MIN}(T_{\text{3D},k}(t - \Delta t), T_{\text{3D},k}(t))$.

- In our current model approach, the thermodynamic plume evolution and microphysics are calculated independently for each trajectory and Bier et al. (2022) find that such an ensemble approach without considering mixing effects among nearby trajectories is not perfect, in particular when the plume is sampled with many trajectories. Hence, we reduce the number of trajectories to $ntr_{\text{sub}}$ by merging $ntr_{\text{gr}} = 25000/ntr_{\text{sub}}$ trajectories that are initially close to each other into a single trajectory. Thereby, we apply a mass-conserving average (that is described more detailed in the supplement of Bier et al. (2022)) to obtain the temporal evolution of the new passive tracer temperature

$$\hat{T}_{\text{3D}}(t) = \sum_{k=1}^{ntr_{\text{gr}}} \frac{T_{\text{3D,k}}(t)}{T_{\text{3D,k}}(t) - T_{\text{3D,a}}}. \tag{3}$$

where $1/(T_{\text{3D,k}}(t) - T_{\text{3D,a}})$ is the weighting factor for a mass-conserving averaging and the equation is exemplarily written for one of the $ntr_{\text{sub}}$ trajectories.



We have performed sensitivity studies for different $ntr_\mathrm{sub}$-values. We find that $ntr_\mathrm{sub} = 1000$ is a reasonable value which still sufficiently resolves the plume hetereogeneity and will be used for the analysis in the present paper.

### 3.3 Contrail formation pathway on entrained ambient particles

In our study, we assume $H_2$ combustion with soot-free emissions. Moreover, we exclude the potential formation of ultrafine particles due to lubricant oil vapor, which will be seperately discussed in Sect. 5. Hence, contrails will solely form on ambient background particles at suitable atmospheric conditions (e.g., Kärcher et al., 1996, 2015). The microphysics of contrail formation on soot particles, as implemented in the LCM, has been described in detail by Bier et al. (2022). These microphysical processes are mostly also relevant for contrail formation on ambient particles and will be summarized briefly in this section. We focus our description on additional aspects for $H_2$ contrails and the associated extensions in the LCM based box model. The major difference besides the chemical composition is the fact that ambient particles are continuously entrained instead of releasing a fixed number of emitted soot particles. Finally, we introduce an alternative activation criterion and an extended homogeneous freezing parameterization. This accounts for the better solubility of the majority of UT particles compared to engine soot.

#### 3.3.1 Ambient aerosol particle properties

We prescribe background particles as an ensemble that is characterized by a log-normal size distribution with a geometric mean dry radius ($\bar{r}_\mathrm{d}$), geometric width ($\sigma_\mathrm{aer}$) and a number concentration ($n_\mathrm{aer}$). Moreover, we specify the hygroscopicity parameter (Petters and Kreidenweis, 2007). The parameters are varied mostly independently of each other representing different types of ambient aerosol particles and accounting for their natural variability.

According to the classical microphysical pathway of contrail formation with the liquid transition phase (e.g., Kärcher et al., 2015; Bier et al., 2022), we consider only CCN or partially soluble mixed particles (i. e., an un-soluble core and a hydrophylic coating), where we will assign the latter simply to "weakly soluble particles". We exclude heterogeneous ice nucleation on insoluble particles since measured and modeled number concentrations of ice nuclei (IN) in the UT are typically several orders of magnitudes lower than those of CCN (e.g., Rogers et al., 1998; Beer et al., 2022). Even though IN may have important effects on natural cirrus cloud properties (e.g., Hendricks et al., 2011), we expect a negligible contribution to the overall contrail ice crystal formation.

For simplicity, we assume that the entrained ambient particles are initially completely dry, i. e., without any environmental hygroscopic water uptake before. We allow for condensational growth of the entrained aerosol particles not only at water-supersaturated conditions but also at plume relative humidities lying between the deliquescence relative humidity ($DRH$) and water saturation. Thereby, $DRH$ is the minimum threshold relative humidity that allows for hygroscopic water uptake by a given substance.





### 3.3.2 Entrainment of ambient particles

Aerosol particles from the environment are continuously entrained into the plume. The following subscripts "E" and "a" denote conditions at the engine exit and in the atmospheric background, respectively. The plume expands over time and its area ($A$) increases according to Bier et al. (2022) as

$$A(t) = A_{\mathrm{E}} \cdot \frac{T(t)}{T_{\mathrm{E}}\,\mathcal{D}(t)}, \tag{4}$$

where $\mathcal{D}(t) = \mathcal{C}_{\mathrm{E}}/\mathcal{C}(t)$ is the dilution factor, which is defined as the ratio between the air-to-fuel ratio ($\mathcal{C}$) at the engine exit plane and at a certain plume age $t$, respectively. We will simply call $\mathcal{C}$ "dilution". With $T$ representing the current plume temperature, the term $\frac{T(t)}{T_{\mathrm{E}}}$ accounts for the temporal change in the plume air density.

The number of aerosol particles (here per flight distance) that have been entrained increases and is calculated according to Kärcher et al. (2015) as

$$N_{\mathrm{aer}}(t) = T_{\mathrm{a}} \left( \frac{A(t)}{T(t)} - \frac{A_{\mathrm{E}}}{T_{\mathrm{E}}} \right) n_{\mathrm{aer}} = \frac{T_{\mathrm{a}}\,A_{\mathrm{E}}}{T_{\mathrm{E}}} \left( \mathcal{D}(t)^{-1} - 1 \right) := \alpha(t)\,n_{\mathrm{aer}}, \tag{5}$$

where $n_{\mathrm{aer}}$ is the aerosol background number concentration. We assume that the initial plume is void of any aerosol particles, which implies that any aerosol particle sucked into the aircraft engine is destroyed. If we prescribed aerosol particles also in the initial plume (with identical properties as in the environment), results would not change too much as their contribution to the total aerosol particle number gets smaller and smaller while the plume expands.

The number of particles being entrained into the plume during one time step $\Delta t$ is then given by

$$\Delta N_{\mathrm{aer}}(t) = (\alpha(t) - \alpha(t - \Delta t))\,n_{\mathrm{aer}}. \tag{6}$$

In every time step, a new SIP ensemble representing these newly entrained particles is created. Only in the initial stages of the simulation when $RH_{\mathrm{wat}}$ is still below $DRH$ no new SIPs have to be created as aerosol particles inside the plume are still dry. In this case, it is sufficient to only increase the SIP weight (i. e., the number denoting how many real particles are represented by a SIP).

Clearly, the continuous creation of new SIPs would cause huge values of $N_{\mathrm{SIP}}$ and lead to computationally expensive or even unfeasible simulations. Hence, we apply a SIP merging algorithm if the overall SIP number $N_{\mathrm{SIP}}$ gets too large (see Appendix A).

Moreover, we found that $T_{\mathrm{3D}}$ happens to increase in certain (short) segments along several plume trajectories. This implies that the cross-sectional area represented by the trajectory shrinks (i.e. $\mathcal{C}(t) <= \mathcal{C}(t - \Delta t)$) and a negative value of $\Delta N_{\mathrm{aer}}(t)$ follows. To inhibit such an unwanted detrainment of particles and hydrometeors, we have smoothed our trajectory data such that the dilution $C(t)$ is a monotonically increasing function with time (see Sect. 3.2).



### 3.3.3 Diffusional growth and freezing

For spherical droplets, the single droplet mass growth equation is given by (Kulmala, 1993)

$$
\frac{dm_{\mathrm{w}}}{dt} = \frac{4\pi r(e_{\mathrm{v}} - e_{\mathrm{K,wat}})}{\frac{R_v T}{D_{\mathrm{v}}}\beta_{\mathrm{m}}^{-1} + \frac{e_{\mathrm{K,wat}} L_{\mathrm{c}}^2}{R_v K T^2}\beta_{\mathrm{t}}^{-1}}, \tag{7}
$$

where $r$ is the wet aerosol or droplet radius, $e_{\mathrm{v}}$ is the partial vapor pressure. $L_{\mathrm{c}}$ denotes the specific latent heat for condensation/evaporation, $D_{\mathrm{v}}$ the binary diffusion coefficient of air and water vapor, $K$ the conductivity of air, $R_{\mathrm{v}}$ the specific gas constant of vapor and $T$ the temperature. The transitional correction factors $\beta_{\mathrm{m}}$ and $\beta_{\mathrm{t}}$ are calculated according to Eqs. (A4) and (A5) of Bier et al. (2022) based on Fuchs and Sutugin (1971). Note that there is a transcription error in the previous study and the denominators of Eqs. (A4) and (A5) miss both the term "+ 1". With this correction, $\beta_{\mathrm{m}}$ and $\beta_{\mathrm{t}}$ tend to one for small Knudsen numbers, as intended. The quantity $e_{\mathrm{K,wat}}$ is the product of the saturation vapor pressure over a flat water surface $e_{\mathrm{sat,wat}}$ and the equilibrium saturation ratio over a solution droplet surface $S_{\mathrm{K}}$. As in Bier et al. (2022), we calculate $S_{\mathrm{K}}$ using the $\kappa$-Köhler Eq. (Petters and Kreidenweis, 2007)

$$
S_{\mathrm{K}} = \frac{r^3 - r_{\mathrm{d}}^3}{r^3 - r_{\mathrm{d}}^3(1 - \kappa)} \cdot \exp\left(\frac{2\hat{\sigma} M_{\mathrm{wat}}}{R T \rho_{\mathrm{wat}} r}\right), \tag{8}
$$

where the first term is the activity of water ($a_{\mathrm{wat}}$) and the exponential expression the Kelvin term. $r_{\mathrm{d}}$ is the particle dry radius, $\kappa$ the hygroscopicity parameter, $\hat{\sigma}$ the surface tension of the solution droplet, $\rho_{\mathrm{wat}}$ the mass density of water, $M_{\mathrm{wat}}$ the molar mass of water and $R$ the universal gas constant. The surface tension typically increases with decreasing $a_{\mathrm{wat}}$ for salt solutions due to negative adsorption and increases with decreasing $a_{\mathrm{wat}}$ for acidic solutions due to positive adsorption. Since we do not prescribe specific aerosol particle species but only the hygroscopicity parameter in the present study, we approximate $\hat{\sigma}$ with the surface tension of pure water droplets and use the polynomial expression by Hacker (1951) as in Bier et al. (2022). Hence, a slight error is caused in the Kelvin term for more concentrated solution droplets.

The entrained ambient aerosol grow by condensation if $RH_{\mathrm{wat}}$ is larger than the deliquescence relative humidity ($DRH$). As long as the entrained particle is completely dry, $S_{\mathrm{K}}$ is not applicable since $r = r_{\mathrm{d}}$. Therefore, we set $S_{\mathrm{K}}$ to a value that is slightly lower than $RH_{\mathrm{wat}}$ for the first time step with condensation. This causes the particle to grow hygroscopically so that $r > r_{\mathrm{d}}$ in the next time step and then $S_{\mathrm{K}}$ is calculated according to Eq. (8). During the subsequent plume evolution, the wetted particle/droplet grows further due to condensation if $RH_{\mathrm{wat}} > S_{\mathrm{K}}$ or shrink due to evaporation if $RH_{\mathrm{wat}} < S_{\mathrm{K}}$ according to Eq. (7).

In Bier et al. (2022), the soot particles have been considered to be activated into water droplets if the wet radius has exceeded the critical radius, which is the radius at the maximum of $S_{\mathrm{K}}$. In the present study, we consider aerosol particles to be activated into water droplets if the activity of water has exceeded a critical value $a_{\mathrm{wat,c}} = 0.90$ to ensure a sufficient water uptake for freezing. Once an aerosol particle has been activated into a droplet, it can freeze to an ice crystal if the plume temperature drops below the homogeneous freezing temperature of that solution droplet ($T_{\mathrm{frz}}$). For the calculation of $T_{\mathrm{frz}}$, Bier et al. (2022) follow the approach of Kärcher et al. (2015) and Riechers et al. (2013) assuming pure water droplets. We extend this approach



by including a simple correction term, based on the parameterization of O and Wood (2016), to account for the decrease in $T_{\mathrm{frz}}$ due to the solution effect (e.g., Koop et al., 2000). Further details are described in Appendix B.

The depositional growth of the formed ice crystals is calculated according to Eq. (7) of Bier et al. (2022) which is based on Mason (1971). Note that there is a transcription error in that equation, where the correction term $\beta_{\mathrm{v}}^{-1}$ occurs twice in the mass 315 diffusion term and should be removed in the denominator.

### 3.4 Model settings and baseline parameters

Tab. 2 summarizes our baseline initial and background conditions for $H_2$ combustion as well as the ambient particle properties and model set-up parameters. We prescribe an ambient temperature $T_{\mathrm{a}}$ of 225 K, ambient pressure $p_{\mathrm{a}}$ of 260 hPa and relative humidity over ice $RH_{\mathrm{ice,a}}$ of 120%. We define a water vapor mass emission index $EI_{\mathrm{v}}$ and specific combustion heat $Q$ that 320 are typical of hydrogen propulsion (Tab. 1). We set the propulsion efficiency, engine exit temperature and initial plume area to the same values as in Bier et al. (2022). The fuel and engine parameters $EI_{\mathrm{v}}$, $Q$, $\eta$ and $T_{\mathrm{E}}$ are kept constant for all sensitivity studies even though they can slightly change with ambient conditions. We determine the initial plume dilution $\mathcal{C}_{\mathrm{E}}$ and the fuel consumption $m_{\mathrm{F}}$ according to Eqs. (9) and (10), which are given in Sect. 5.2. Since $A_{\mathrm{E}}$ is the area of one engine nozzle exit plane (based on the FLUDILES data for the A340-300 aircraft and kept constant in this study), our $m_{\mathrm{F}}$-value is representative 325 of a single engine and would be 4 times larger for the whole aircraft. Thus, we simulate contrail formation behind a single aircraft engine. The $\mathcal{C}_{\mathrm{E}}$ and $m_{\mathrm{F}}$-values listed in Tab. 1 are given for the atmospheric baseline $T_{\mathrm{a}}$ and $p_{\mathrm{a}}$-values. Note that $\mathcal{C}_{\mathrm{E}}$ is larger and $m_{\mathrm{F}}$ lower by a factor of 2.8 than for kerosene combustion because $Q$ is accordingly higher (with the relations $\mathcal{C}_{\mathrm{E}} \sim Q$ and $m_{\mathrm{F}} \sim Q^{-1}$).

We prescribe ambient particles with a mono-modal log-normal size-distribution. We set the geometric-mean dry radius ($\bar{r}_{\mathrm{d}}$) 330 to 15 nm and geometric width of 1.6 representing a typical Aitken aerosol mode in the UT (e.g., Brock et al., 2021). We define an aerosol number concentration ($n_{\mathrm{aer}}$) of 600 cm$^{-3}$, which is lying well in between the observed values for Aitken and accumulation mode particles (e.g., Minikin et al., 2003; Borrmann et al., 2010). We set the hygroscopicity parameter $\kappa$ to 0.5, which is e. g. a typical value of ammonium sulfate particles (Liu et al., 2014). Peng et al. (2022) provide a large data set for $DRH$ of different atmospheric compounds (e.g. see their Table 1). Even though many inorganic compounds have a 335 $DRH$ significanlty below one, we set our baseline $DRH$ very close to water saturation. The reason for that will be discussed in Sect. 5.2. For a comparison with conventional contrails, we also provide associated soot particle properties in Tab. 2.

For any ambient particle ensemble, we use around 110 SIPs to represent its log-normal size distribution. For this, we use the algorithm described in Unterstrasser and Sölch (2014), which has favorable numerical convergence properties (Unterstrasser et al., 2017b). Each simulation contains an ensemble of box model runs for 1000 different trajectory data as described in the 340 previous section. The runs are performed independently of each other for each trajectory. The standard simulation time is 3 s; the numerical time step is 0.001 s. For some simulations with $T_{\mathrm{a}} < 220$ K, we extend the simulation time to 5 s since the time period where droplet and, hence, ice crystal formation occurs is longer than 3 s.





| ambient conditions | $H_2$ fuel/engine prop. | $H_2$ engine exit conditions | ambient particle prop. | soot part. prop | set-up parameters |
|---|---|---|---|---|---|
| $T_a = 225\,\mathrm{K}$ | $EI_v = 8.94\,\mathrm{kg\,kg^{-1}}$ | $T_E = 580\,\mathrm{K}$ | $n_{aer} = 600\,\mathrm{cm^{-3}}$ | $N_s = 1.6\cdot10^{12}\,\mathrm{m^{-1}}$ | $t_{sim} = 3\text{--}5\,\mathrm{s}$ |
| $p_a = 260\,\mathrm{hPa}$ | $Q = 1.2\cdot10^8\,\mathrm{J\,kg^{-1}}$ | $A_E = 0.25\pi\,\mathrm{m^2}$ | $\bar{r}_d = 15\,\mathrm{nm}$ | $\bar{r}_d = 15\,\mathrm{nm}$ | $dt = 0.001\,\mathrm{s}$ |
| $RH_{ice,a} = 120\%$ | $\eta = 0.36$ | $\mathcal{C}_E \approx 210$ | $\sigma_s = 1.6$ | $\sigma_{aer} = 1.6$ | $dt_{out} = 0.01\,\mathrm{s}$ |
| — | — | $m_F \approx 0.58\,\mathrm{g\,m^{-1}}$ | $\kappa = 0.5$ | $\kappa = 0.005$ | $N_{SIP,0} \approx 110$ |
| — | — | — | $DRH = 0.99$ | $DRH = 0.99$ | $N_{SIP,m} = 1600$ |

**Table 2.** Baseline parameters for our LCM box model studies. The fuel, engine properties and exit conditions refer to $H_2$ combustion. In addition to the entrained ambient particle properties, we provide the baseline soot particle properties for a comparison of $H_2$ with conventional contrails.

## 4 Results

In this section, we first analyze the temporal evolution of thermodynamic and microphysical $H_2$ contrail properties for our
baseline case (Sect. 4.1). We then investigate the impact of atmospheric conditions on those properties in Sect. 4.2. In Sect. 4.3, we analyze the influence of ambient aerosol particle properties on contrail ice crystal formation either prescribing one or two co-existing aerosol particle ensembles. Finally, we compare our results with conventional kerosene contrails in terms of ice crystal number and optical thickness in Sect. 4.4. The thermodynamic and microphysical properties are either averaged or summed up over all box model trajectories obeying a mass-conserving weighting as each trajectory may represent a different
share of the plume at later times. In this study, we always display our microphysical properties in units (number or mass) per flight distance.

### 4.1 Temporal evolution of contrail properties for the baseline case

Fig. 2 shows the temporal evolution of thermodynamic and microphysical properties for our baseline case defined in Tab. 2. The mean plume temperature (panel (a)) decreases with increasing plume age due to continuous mixing of the exhaust with ambient
air approaching the ambient temperature. Accordingly, the plume dilution and, therefore, plume area increase, the latter from around 1 to $280\,\mathrm{m^2}$ after 3 s. The mean relative humidity over water $RH_{wat}$ (panel (b)) surpasses the deliquescence relative humidity after around 0.2 s and reaches its maximum of around 220% after 0.4 s. Compared to a plume behind a conventional aircraft (e.g., see Fig. 2 in Bier et al. (2022)), our maximum $RH_{wat}$ and accordingly $RH_{ice}$ values are substantially higher since the difference between the ambient and the SA-threshold temperature is larger (the SA-threshold temperature is by around
10 K higher for $H_2$ than for kerosene combustion, see Sect. 2.2).

Panel (c) shows the accumulated number of aerosol particles entrained into the plume ($N_{aer}$), the number of formed droplets ($N_{drp}$) and the number of ice crystals ($N_{ice}$). $N_{aer}$ increases nearly linearly with time reaching values of around $6\cdot10^{10}$ and $1.6\cdot10^{11}\,\mathrm{m^{-1}}$ after 1.5 and 3 s, respectively. This is different to exhaust species like soot particles since their total number in the plume does not depend on the dilution state. The $N_{aer}$-values are mainly controlled by the ambient aerosol number





concentration ($n_\mathrm{aer}$) and the plume area expansion. The first aerosol particles activate into water droplets (dashed line) after $RH_\mathrm{wat}$ surpasses the $DRH$ in the corresponding trajectories. A few tenth of seconds later, they freeze to ice crystals (solid line) once plume temperature falls below the homogeneous freezing temperature of those droplets. Later on (at plume ages between around 0.6–1.1 s), $N_\mathrm{ice}$ is very close to $N_\mathrm{aer}$. This means that basically all entrained aerosol particles nearly instantaneously form droplets and freeze. After 1.5 s when the mean $RH_\mathrm{wat}$ falls below approximately 95%, no further droplets and ice crystals

form. Therefore, $N_\mathrm{ice}$ stays constant afterwards at $5.4 \cdot 10^{10}$ m$^{-1}$. Some of the droplets (small peak in $N_\mathrm{drp}$ at around 1.4 s) cannot freeze and evaporate afterwards. This is because the homogeneous freezing temperature of those droplets, which are either too small or have a too low water activity, is not reached. The ice crystals grow by deposition so that the ice water mass ($m_\mathrm{ice}$) continuously increases (panel (d)). The mean ice crystal radius $\overline{r}_\mathrm{ice}$ (red line) tends to increase over time and finally reaches a size of around 2.2 $\mu$m. The decrease of $\overline{r}_\mathrm{ice}$ between 0.6 and 1.3 s is because more and more of the smaller

droplets manage to freeze to ice crystals and, hence, the mean ice crystal size drops in that time period. Since the plume is still ice-supersaturated (solid line in panel (b)) after 3 s, $m_\mathrm{ice}$ and $\overline{r}_\mathrm{ice}$ would increase even further.

## 4.2 Impact of atmospheric properties

Here, we investigate in detail the impact of ambient temperature on the plume thermodynamical and microphysical contrail properties. Moreover, we analyze the influence of ambient pressure and relative humidity over ice. We prescribe our baseline

ambient aerosol particle properties and keep them constant in this section. Note that a fixed aerosol number concentration for varying atmospheric parameters (in particular pressure) is an idealized assumption in the following sensitivity studies.

### 4.2.1 Influence of ambient temperature on temporal evolution of thermodynamic and contrail properties

Fig. 3 highlights the strong impact of ambient temperature $T_\mathrm{a}$ on contrail ice crystal formation. As shown in panel (a), the plume temperature at a given plume age is certainly lower in a colder environment and finally approaches the corresponding

$T_\mathrm{a}$ value. The peak mean relative humidity over water (displayed in panel (b)) increases with decreasing $T_\mathrm{a}$ (reaching values of around 350% and 550% for $T_\mathrm{a}$ of 220 K and 215 K, respectively). The large increase for low $T_\mathrm{a}$-values is due to the non-linearity between saturation vapor pressure and temperature. The relative humidity over ice behaves accordingly (not shown). The slight change in the evolution of $N_\mathrm{aer}$ is a consequence of our model set-up with a fixed aerosol number concentration and the varying plume air density with temperature (at fixed ambient pressure). In general, the droplet formation is basically

controlled by the time period where $RH_\mathrm{wat}$ is above $DRH$ such that water can condense on the entrained aerosol particles. (Note that the evolution in $RH_\mathrm{wat}$ and, therefore, this time period varies with each trajectory and we here only display the ensemble mean quantity.) This mean time period for possible droplet and ice crystal formation substantially increases with decreasing ambient temperature (e.g. from 0.15 s up to 2.8 s for $T_\mathrm{a} = 215$ K). This means that ice crystal formation is initiated earlier and comes to a halt later (see red and blue solid lines in panel (c)) compared to the baseline case. Moreover, nearly all

formed droplets freeze very quickly to ice crystals so that $N_\mathrm{drp}$ approaches zero. For these reasons, the final ice crystal number strongly increases with decreasing $T_\mathrm{a}$ over the whole temperature range (see also Fig. 4). This is different to what we find for conventional soot contrails as all soot particles turn into ice crystals if $T_\mathrm{a}$ is several K below the SA-threshold temperature





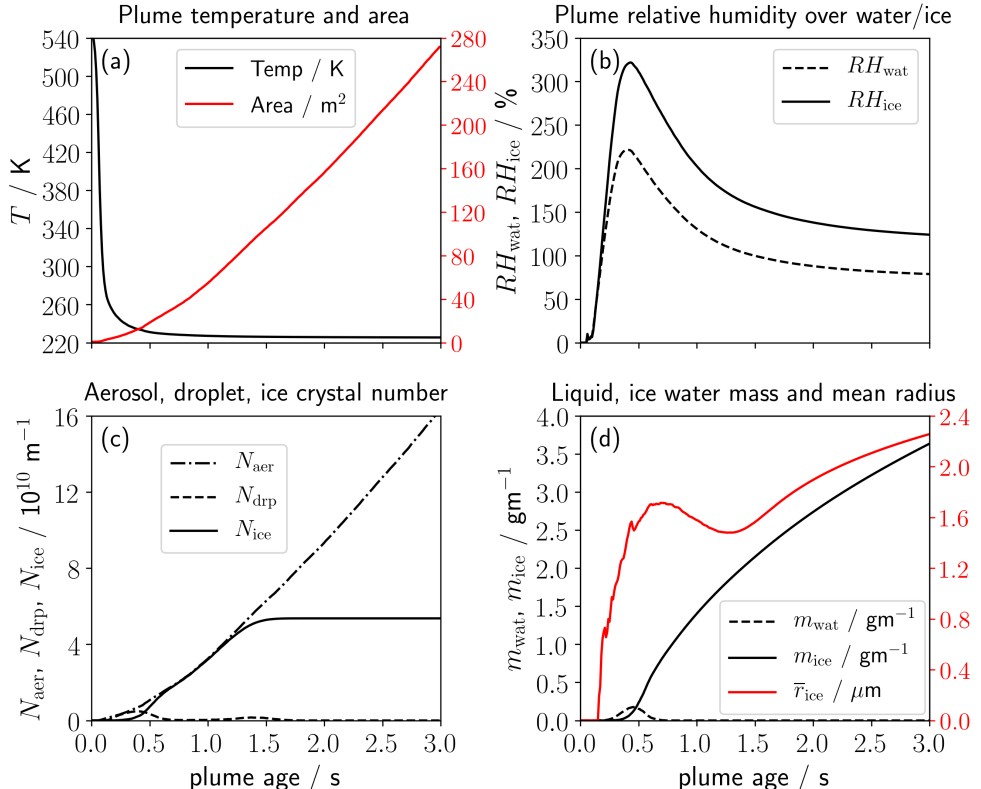

**Figure 2.** Temporal evolution of thermodynamic and microphysical properties in the plume for the baseline case: The panels show (a) temperature (black) and cross-sectional area (red and labels on the right axis), (b) relative humidity over water (dashed) and ice (solid), (c) number of aerosol particle entrained into the plume (dash-dotted), number of droplets (dashed) and ice crystals (solid) per flight distance and (d) liquid (black dashed) and ice water mass per flight distance (black solid) and mean radius of the ice crystals (red and labels on the right axis).

(e.g., Kärcher et al., 2015; Bier and Burkhardt, 2019; Bier et al., 2022). Yet, any further reduction in $T_a$ does not lead to more ice crystals in the conventional case. Moreover, the peak plume $RH_{wat}$-values are substantially higher for $H_2$ than for kerosene

combustion at same ambient conditions. Therefore, droplet and ice crystal formation on ambient particles is controlled more strongly by the time period in which the plume is water-supersaturated than by the maximum water-supersaturation.

For higher ambient temperatures ($T_a >= 230$ K), many droplets cannot freeze to ice crystals and evaporate thereafter (which is indicated by declining $N_{drp}$ at nearly constant $N_{ice}$). This is because the homogeneous freezing temperature of the smaller and/or more concentrated solution droplets is below the plume/ambient temperature. Hence, the final ice crystal numbers are

decreased further in addition to the fact that the time period for possible droplet formation is lower. The decrease of $N_{ice}$ with increasing $T_a$ becomes stronger for $T_a >= 232$ K (see also Fig. 4) and for $T_a = 233$ K, only a few large droplets can form ice crystals. For higher ambient temperatures no ice crystal formation occurs anymore. This means that for $H_2$ combustion, the



freezing temperature is typically smaller than the SA-threshold temperature and becomes a more limiting criterion for contrail formation. Yet, the SA-threshold temperature is still relevant, as its difference to the ambient temperature determines the peak

and time period of water supersaturation in the plume. The ice water mass (shown in panel (d) in general increases with decreasing ambient temperature. The strong increase between $T_a$ of 230 K and 233 K is mainly due to the increase in $N_{ice}$.

In the following sections, we will focus our analysis on the final number of formed ice crystals ($N_{ice,f}$) since the young contrail ice number mostly impacts the further contrail (cirrus) properties and radiative forcing (e.g., Bier et al., 2017; Burkhardt

et al., 2018; Bier and Burkhardt, 2022). In contrast, the initial ice water mass/mean ice crystal radius was shown to have a low impact on the contrail life cycle in the dispersion phase (e.g., Unterstrasser and Gierens, 2010), but the size distribution of the formed contrail ice crystals can strongly impact the sublimation loss of ice crystals during the vortex phase (e.g., Unterstrasser, 2014). The latter will be investigated in future studies.

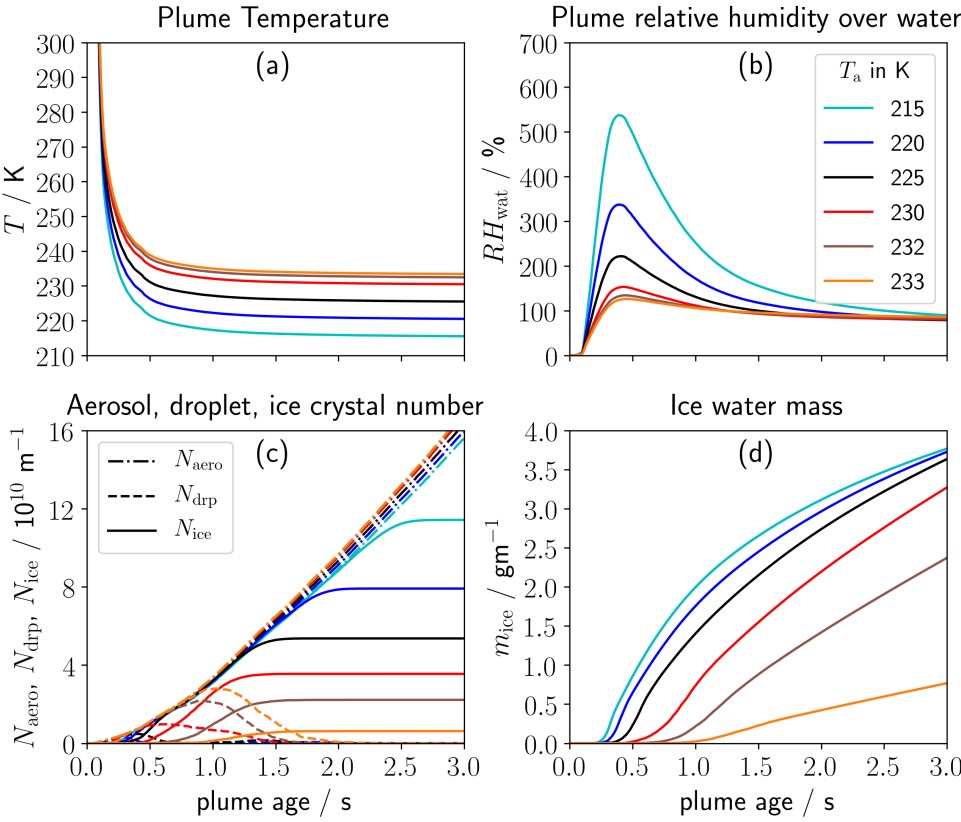

**Figure 3.** Impact of ambient temperature ($T_a$) on temporal evolution of thermodynamic and microphysical properties in the plume: The panels show (a) temperature, (b) relative humidity over ice, (c) number of aerosol particle entrained into the plume (dash-dotted), number of droplets (dashed) and ice crystals (solid) per flight distance and (d) ice water mass per flight distance. The colors represent the different $T_a$-values as defined in the legend.





### 4.2.2 Final ice crystal number

Fig. 4 displays $N_{ice,f}$ versus ambient temperature $T_a$ for (a) 3 different pressure $p_a$ and (b) 3 different ambient relative humidity over ice $RH_{ice,a}$-values. Note that our parameter settings are simplified in the sense that some combinations of paramter-values are not realistic for the atmosphere (e.g., lowest $T_a$ value at the highest $p_a$ value). Compared to the previous subsection, we now include a further case with $T_a = 210$ K (that requires a longer simulation time since the period where the plume is water-supersaturated is higher than 3 s). This case emphasizes the enhanced increase of $N_{ice,f}$ with decreasing $T_a$ for very cold

conditions and is consistent with the findings by Ström and Gierens (2002). $N_{ice,f}$ is increased for a higher ambient pressure because the slope of the mixing line $G$, defined by Eq. (1), is larger for a higher pressure. Moreover, panel (b) shows an incline of $N_{ice,f}$ with increasing $RH_{ice,a}$. Both the increased $G$ and the higher $RH_{ice,a}$ lead to higher peak plume relative humidities and enlarge the time period for possible droplet and subsequent ice crystal formation. Interestingly, this increase is quite strongly pronounced for the highly ice-supersaturated case (red line) at ambient temperatures between 230 and 234 K (and for 234 K

ice crystals can only form for this case at all). This is because the droplet freezing is mainly limited by the droplet size in that $T_a$-range and for the high $RH_{ice,a}$-case, more larger droplets can form that turn into ice crystals.

Finally, ambient temperature is the parameter that influences the number of formed contrail ice crystals most while the impact of ambient pressure and relative humidity is clearly smaller. This behavior is similar to the conventional case with kerosene combustion.

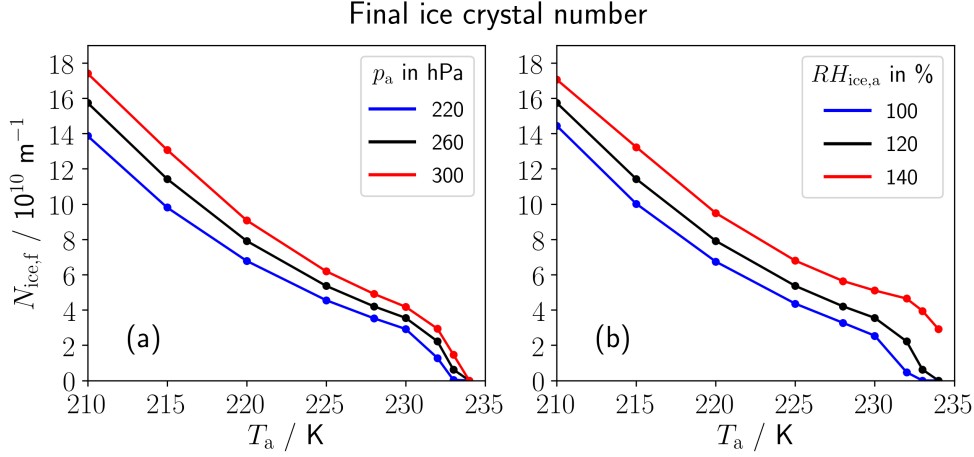

**Figure 4.** Final ice crystal number per flight distance ($N_{ice,f}$) versus ambient temperature for (a) three different ambient pressures ($p_a$) and (b) for three different ambient relative humidities over ice ($RH_{ice,a}$). The black line in both panels always refers to the baseline $p_a$ and $RH_{ice,a}$-values. $N_{ice,f}$ is given at a plume age of 3 s for $T_a >= 215$ K, and at a plume age of 5 s for $T_a = 210$ K.





### 4.3 Sensitivity of ice crystal number to ambient aerosol particle properties

In this section, we investigate the impact of ambient aerosol particle properties on the (final) number of formed ice crystals in $H_2$ contrails. We prescribe one aerosol particle ensemble (single mode) in Sect. 4.3.1 (as in the previous analysis) and two co-existing aerosol particle ensembles in Sect. 4.3.2.

#### 4.3.1 Studies with a single aerosol particle ensemble

Fig. 5 shows the variation of $N_{\mathrm{ice,f}}$ with different aerosol particle properties. The sensitivities are always shown for three ambient temperatures $T_a$ (differentiated by the color). In general, we see an increase in $N_{\mathrm{ice,f}}$ with decreasing $T_a$ for any particle property combination, being consistent with the findings in the previous section. Panel (a) shows that $N_{\mathrm{ice,f}}$ inclines with increasing aerosol number concentration $n_{\mathrm{aer}}$. This increase becomes weaker for higher $n_{\mathrm{aer}}$ ($> \approx 200\,\mathrm{cm}^{-3}$) values. This is due to the enhanced competition for plume water vapor between the growing droplets/ice crystals for increased aerosol number concentrations.

Next, we analyse the importance of the mean dry radius of the aerosol size distribution $\overline{r}_{\mathrm{d}}$ and the hygroscopicity parameter $\kappa$. Typically, the ice crystal number strongly increases with increasing mean dry size for $\overline{r}_{\mathrm{d}} < \approx 10\,\mathrm{nm}$ and then stays nearly constant (panel (b)). The increase is mainly due to the Kelvin effect, i. e., larger aerosol particles are easier to activate into water droplets since they require lower plume water-supersaturations and grow more quickly to water droplets (e.g., Bier et al., 2022).

Panel (c) shows the dependence of $N_{\mathrm{ice,f}}$ on $\kappa$. For $T_a = 230\,\mathrm{K}$, $N_{\mathrm{ice,f}}$ slightly increases for a larger $\kappa$ for all three $\overline{r}_{\mathrm{d}}$ as indicated in the legend. For the lower $T_a$ cases, the variation of $N_{\mathrm{ice,f}}$ with $\overline{r}_{\mathrm{d}}$ and $\kappa$ is more complex. For the small sized particles (dotted-dashed lines), there are two counteracting effects: $N_{\mathrm{ice,f}}$ increases with rising hygroscopicity parameter for $\kappa < 0.1$ since better soluble particles can easier form water droplets. On the other hand, $N_{\mathrm{ice,f}}$ subsequently decreases. This is because some of the droplets cannot freeze to ice crystals since their water activity is lower due to the enhanced solution effect for higher $\kappa$ and, therefore, the homogeneous freezing temperature is significantly decreased (see Fig. B1 in the appendix). For the other $\overline{r}_{\mathrm{d}}$ cases, $N_{\mathrm{ice,f}}$ hardly changes with the solubility and mean aerosol particle size.

We also analyze in panel (d) the impact of the geometric width of the aerosol size distribution for four different $\overline{r}_{\mathrm{d}}$, $\kappa$-combinations as displayed in the legend. Our results imply a very low sensitivity of $N_{\mathrm{ice,f}}$ to the geometric width.

In conclusions, the sensitivity of the ice crystal number to $\overline{r}_{\mathrm{d}}$ and $\kappa$ is low for aerosol particles with a large mean dry size. For the small-sized particles, we find a quite complex variation of $N_{\mathrm{ice,f}}$ with $\kappa$, mainly for low ambient temperatures, due to various counteracting effects.




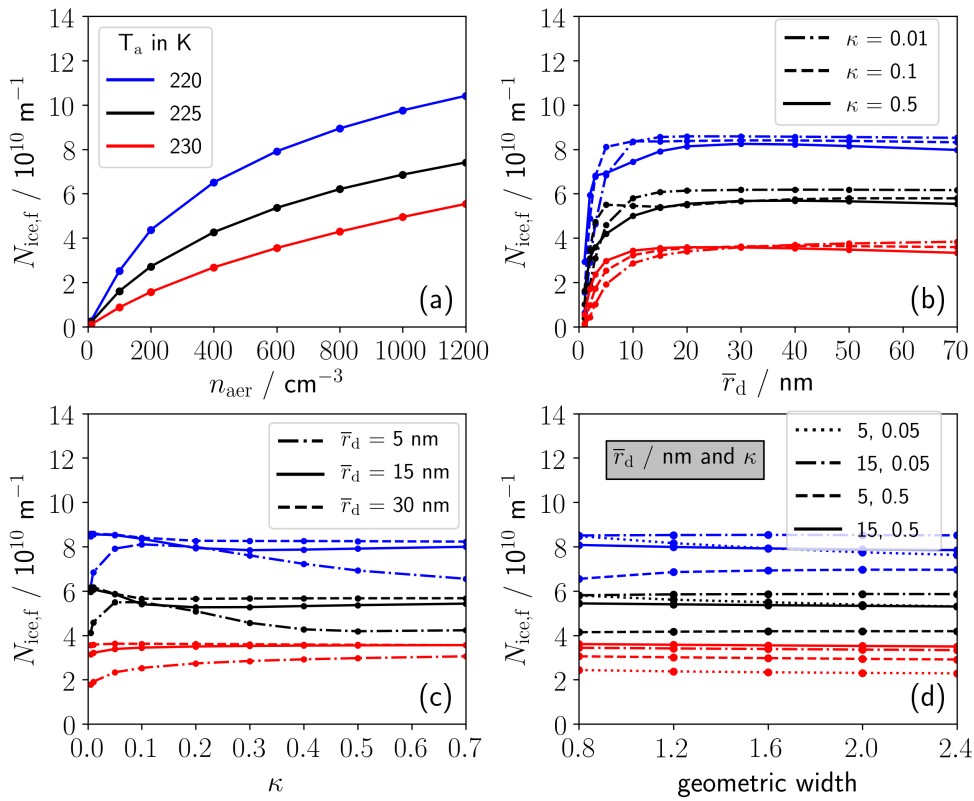

**Figure 5.** Final ice crystal number per flight distance ($N_{\text{ice,f}}$) depending on different aerosol particle properties assuming uni-modal size distributions for three ambient temperatures ($T_a$) (different colors defined in legend (a)): $N_{\text{ice,f}}$ is shown versus (a) ambient aerosol number concentration, (b) geometric mean dry radius ($\overline{r}_d$) for three solubility-values (see line style in legend (b)), (c) hygroscopicity parameter ($\kappa$) for three $\overline{r}_d$-values (see line style in legend (c)) and (d) the geometric width of the size distribution for 4 different $\overline{r}_d$, $\kappa$ combinations as displayed in the legend.





### 4.3.2  Studies with two co-existing aerosol particle ensembles

So far, the aerosol particles were prescribed with a single log-normal size distribution and a fixed hygroscopicity value. In the
present section, we prescribe two co-existing ambient aerosol particle ensembles and analyze contrail ice crystal formation
for two ambient temperatures. The given $n_{aer}$ value is the total number concentration of both aerosol ensembles. We restrict
our analysis to cases where each ensemble has a number concentration of $0.5 \cdot n_{aer}$. We consider nucleation mode ($\bar{r}_d = 3$ nm),
Aitken mode ($\bar{r}_d = 15$ nm as in our baseline case) and accumulation mode ($\bar{r}_d = 50$ nm) particles. The given mean dry sizes of
the single modes are prescribed consistently with typical observed UT geometric mean diameters over the Atlantic and Pacific
Ocean within the ATom campaign (see Fig. 12 of Brock et al. (2021)). Moreover, we consider well soluble ($\kappa = 0.5$) particles
like inorganic salts and weakly soluble ambient particle ($\kappa = 0.05$) like organic species or aviation soot. We restrict our analysis
to a scenario, where the two co-existing particle ensembles always differ only in one parameter, either $\bar{r}_d$ or $\kappa$. Moreover, we
compare the total $N_{ice}$ of the two aerosol particle ensembles with a reference case, which uses a single aerosol population with
the same value of $n_{aer}$ and the average $\bar{r}_d$ and $\kappa$-values of the two particle ensembles.

The first two rows of Fig. 6 show the temporal evolution of $N_{ice}$ with a fixed $n_{aer} = 600$ cm$^{-3}$. First, we analyze the contrail
ice number evolution for bi-modal aerosol size distributions with same $\kappa$ (first row). Panel (a) shows that $N_{ice}$ for the Aitken
mode particle is at the end by around 50% larger than for the nucleation mode particle for both temperatures. This is because,
for a given plume relative humidity, the larger particles can better activate into water droplets (and freeze thereafter) than
the very small nucleation mode particles due to the Kelvin effect. Moreover, the total ice crystal number is at the end by
around 10 % lower than the reference $N_{ice}$. Considering the Aitken and accumulation mode in panel (b), $N_{ice}$ between the two
particle ensembles is quite similar and for $T_a = 230$ K nearly identical. This is consistent with the findings for a single particle
ensemble where the variation of the ice crystal number for $\bar{r}_d > \approx 10$ nm is low (see Fig. 5 (b)). For the same reason, the total
and reference $N_{ice}$ are close to each other.

Now, we investigate the ice crystal formation for particle ensembles with two different solubility characteristics but same $\bar{r}_d$
(second row). For $T_a = 230$ K, $N_{ice}$ for the weakly soluble particles is lower than for the well soluble particles. This is because
the less hygroscopic particles are harder to activate and/or cannot grow to sufficiently large droplet sizes in order to freeze to
ice crystals. Hence, the total ice crystal numbers are slightly reduced (by around 5–10%) relative to the reference ice numbers.
For $T_a = 225$ K, $N_{ice}$ is at the end lower for $\kappa = 0.5$ than for $\kappa = 0.05$ and the total ice number is lower than the reference ice
number contrary to the high $T_a$ cases. This is due to the decrease in $T_{frz}$ for droplets with a higher solution effect (lower $a_{wat}$),
as explained in Sect. 4.3.1 and shown in Fig. 5 (c).

The last two rows show the final ice crystal numbers ($N_{ice,f}$) for the same aerosol particle ensembles as in panels (a)–(d) but for
different aerosol number concentrations. Basically, we see a similar trend for the ice crystal numbers of the single ensembles
and the reference case as in the panels above throughout the whole $n_{aer}$-range. In general, the increase in $N_{ice,f}$ with increasing
$n_{aer}$ becomes weaker for higher $n_{aer}$, in particular for the low $T_a$ cases. This is consistent with the findings for a single aerosol
ensemble (as already shown in Fig. 5 (a)). The flattening in $N_{ice,f}$ is pronounced most strongly for the nucleation mode and the
weakly hygroscopic particles. Again, the total $N_{ice,f}$ of the two-particle ensemble is nearly always reduced compared to the



reference $N_{\mathrm{ice,f}}$ of the single average particle ensemble except for the low $T_{\mathrm{a}}$ cases in panel (g) and (h).

Finally, we find largest differences between the total ice crystal number of the co-existing particle ensembles and the associated average single particle ensemble for the cases where nucleation mode particles are involved. This is mainly due to the non-linearity between (final) contrail ice crystal number and mean dry size for those very small aerosol particles.

## 4.4 Comparison of H$_2$ contrails with conventional contrails

In the present section, we study differences in microphysical and optical properties of H$_2$ contrails compared to conventional contrails formed behind aircraft with kerosene combustion. For the latter, we include in our set-up both ice crystal formation on soot and on the entrained ambient particles. We analyze in Sect. 4.4.1 the number of formed contrail ice crystals as a first step to estimate the mitigation potential of H$_2$ combustion. In Sect. 4.4.2, we analyze the optical thickness, which can provide information about the visibility of young contrails. While first measurements like Blue Condor could use this information for their planning, they potentially also provide a first chance for the evaluation of our model. We use the respective engine and fuel parameters defined in Tab. 1. We define the properties of ambient and soot particles according to Tab. 2.

### 4.4.1 Mitigation potential

Fig. 7 compares ice crystal numbers of conventional and H$_2$ contrails. Panel (a) displays the ice crystal number $N_{\mathrm{ice,f}}$ behind a conventional aircraft as a function of ambient temperature and for three typical soot number emission levels. For our calculated fuel consumption (that accounts for the different $Q$-values of H$_2$ and kerosene), the displayed soot particle numbers ($N_{\mathrm{s}} = 0.8 \cdot 10^{12}\,\mathrm{m}^{-1}$, $1.6 \cdot 10^{12}\,\mathrm{m}^{-1}$ and $3.2 \cdot 10^{12}\,\mathrm{m}^{-1}$) represent soot number emission indices of around $5 \cdot 10^{14}$, $10^{15}$ and $2 \cdot 10^{15}\,\mathrm{kg}^{-1}$, respectively. Prescribing our baseline ambient pressure and ambient relative humidity, the SA-threshold temperature $\Theta_{\mathrm{G}}$ for kerosene is around $227\,\mathrm{K}$ ($\approx 10\,\mathrm{K}$ lower than that for H$_2$). Very close to $\Theta_{\mathrm{G}}$, only a few soot and the entrained ambient aerosol particles can form ice crystals due to very low plume water-supersaturations. For ambient temperatures ($T_{\mathrm{a}} \lessgtr \Theta_{\mathrm{G}}$ - 0.5 K), ice crystals mainly form on soot particles since $N_{\mathrm{s}}$ is by around 2 orders of magnitude higher than $N_{\mathrm{aer}}$ during the contrail formation time (not shown). Consistent with previous studies (e.g., Kärcher et al., 2015; Bier and Burkhardt, 2019; Bier et al., 2022), $N_{\mathrm{ice,f}}$ strongly increases with decreasing $T_{\mathrm{a}}$ and then approaches the respective $N_{\mathrm{s}}$-values for sufficiently low $T_{\mathrm{a}}$. For higher $N_{\mathrm{s}}$, the number of formed ice crystals rises more steeply and approaches $N_{\mathrm{s}}$ at lower $T_{\mathrm{a}}$.

Now, we investigate the ratio of the ice crystal numbers between H$_2$ and kerosene contrails ($N_{\mathrm{ice,f,rel}}$) shown in panels (b)–(d). We constrain our analysis to that temperature range in which kerosene contrails are able to form according to the SA-criterion. A mitigation is achieved for $N_{\mathrm{ice,f,rel}} < 1$, where a lower value is connected with a higher mitigation potential. Panel (b) shows $N_{\mathrm{ice,f,rel}}$ versus $n_{\mathrm{aer}}$ for the three $N_{\mathrm{s}}$ (see line style) and for three $T_{\mathrm{a}}$ cases (different colors). In general, we see a clear decrease in $N_{\mathrm{ice,f,rel}}$ with increasing $N_{\mathrm{s}}$ and decreasing $n_{\mathrm{aer}}$. Thereby, $N_{\mathrm{ice,f,rel}}$ is below 0.1 for $n_{\mathrm{aer}} <= 600\,\mathrm{cm}^{-3}$ and below 0.15 for higher $n_{\mathrm{aer}}$. Interestingly, we see for the higher $N_{\mathrm{s}}$ and $n_{\mathrm{aer}}$ cases a larger difference in $N_{\mathrm{ice,f,rel}}$ between 220 and 222 K than between 220 and 225 K.

Therefore, we investigate in more detail the temperature dependency of $N_{\mathrm{ice,f,rel}}$ in the second row of the Figure. Panel (c) shows the relative change in the ice crystal number w. r. t. the three soot cases for our baseline $n_{\mathrm{aer}}$. In connection with the



minimum at $T_a \approx 224\,\mathrm{K}$, there are two different dominating effects: Below 224 K, $N_{\mathrm{ice,f,rel}}$ increases with decreasing $T_a$, in particular for the high soot case. This is because $N_{\mathrm{ice,f}}$ for the kerosene contrails approaches $N_s$ with decreasing $T_a$ (well below the SA-threshold) while $N_{\mathrm{ice,f}}$ for the $H_2$ contrails increases further for lower $T_a$. The latter is because the ambient aerosol is continuously entrained into the plume and the time period for droplet and ice crystal formation increases for colder ambient conditions due to a longer lasting water-supersaturation (see Sect. 4.2). The strong increase in $N_{\mathrm{ice,f,rel}}$ above around 225 K results from the strong decrease of $N_{\mathrm{ice,f}}$ for the conventional contrail. This is because ice crystal formation on the weakly soluble soot particles becomes more and more limited the closer the ambient temperature approaches the SA-threshold temperature. Thereby, $N_{\mathrm{ice,f,rel}}$ is around 0.3 for $T_a = 226.5\,\mathrm{K}$ and around 20 for $T_a = 227\,\mathrm{K}$ (the latter not visible in the figure). Finally, we show the change in ice crystal number relative to our baseline soot case but for three different $n_{\mathrm{aer}}$-values. The sensitivity of $N_{\mathrm{ice,f,rel}}$ to $n_{\mathrm{aer}}$ is quite similar to that to $N_s$ in panel (c). The most obvious difference is that an increase of $n_{\mathrm{aer}}$ by a factor of 2 (brown line in (d)) has a much lower impact on $N_{\mathrm{ice,f,rel}}$ than a decrease in $N_s$ by the same factor (dashed line in panel (c)). This is due to a weaker increase of the $H_2$ contrail ice crystal number with increasing $n_{\mathrm{aer}}$ for high number concentrations (shown in Fig. 5 a).

We can conclude that a switch to $H_2$ combustion indicates a high mitigation potential if the ambient temperature is by more than 0.5 K lower than the SA-threshold temperature for kerosene. This is mainly because $N_{\mathrm{aer}}$ during contrail formation is by around 2 orders of magnitude lower than $N_s$ for typical $n_{\mathrm{aer}}$-values. Another aspect is that the fuel consumption of the cryoplane is in our study by around a factor of 3 lower than that of conventional aircraft. If air traffic with $H_2$ combustion occurs at ambient temperatures between the kerosene SA-threshold temperature and the droplet freezing temperature, clearly additional contrails are produced, which would be absent in the case of kerosene combustion.

### 4.4.2 Contrail visibility

We investigate the young contrail optical thickness ($\tau$) both for kerosene and for $H_2$ combustion. For the quantities analyzed and presented so far, we used a reduced trajectory data set after merging trajectories with similar radial coordinates (see Sect. 3.2). However, the column-wise computation of $\tau$ requires spatial information of the trajectories, namely the lateral and vertical Cartesian coordinates x and z. Hence, the results presented next were obtained by using the full trajectory data set from Vancassel et al. (2014). Fig. 8 (a)–(d) shows the contrail width (indicated by the x-coordinate)-plume age ($t$) distribution of $\tau$: The first row displays the kerosene and the second row our $H_2$ baseline case, both for $T_a$ of 220 K and 225 K, respectively. Observations suggest that the threshold $\tau$ for the visibility of contrails is around 0.05 (e.g., Kärcher et al., 2009). Panel (a) indicates that the kerosene contrail at $T_a = 220\,\mathrm{K}$ becomes visible after around 0.3 s of plume age. Afterwards, the plume quickly spreads and $\tau$ tends to increase due to further formation and growth of ice crystals. Peak values of around one and slightly higher are reached for $t$ between 1 and 1.5 s. For $T_a = 225\,\mathrm{K}$, ice crystals form later and the contrail is visible after 0.6 s. Maximum $\tau$ is by around 50% lower than for the low $T_a$ case since that contrail forms near the formation threshold and the nucleated ice crystal number is significantly reduced (see Fig. 7 (a), black line). For the $H_2$ case, the optical thickness is substantially decreased compared to kerosene contrails, being consistent with the findings by Ström and Gierens (2002). Moreover, these contrails become visible around a half of a second later than those for kerosene at same $T_a$.





Finally, panel (e) shows the temporal evolution of the 90 Percentile optical thickness over the contrail width $\tau_{90}$. We juxtapose
the kerosene contrail with three $H_2$ contrails with different $n_{aer}$-values. Consistent with the spatio-temporal distributions, $\tau_{90}$
of the $H_2$ contrails is significantly lower than for the kerosene contrails (in particular for $T_a = 220\,K$). The optical thickness
decreases for lower $n_{aer}$-values. We expect that the $H_2$ contrail for $n_{aer} = 100\,cm^{-3}$ would be hardly visible. A lower temperature
causes a slight increase in $\tau_{90}$ for the higher $n_{aer}$ cases and for $t < 1.5\,s$. This is mainly due to the higher ice water content for
lower $T_a$ (not shown). Instead, the much stronger increase of $\tau$ for the kerosene case is due to the increased ice crystal number
already at the beginning of contrail formation.

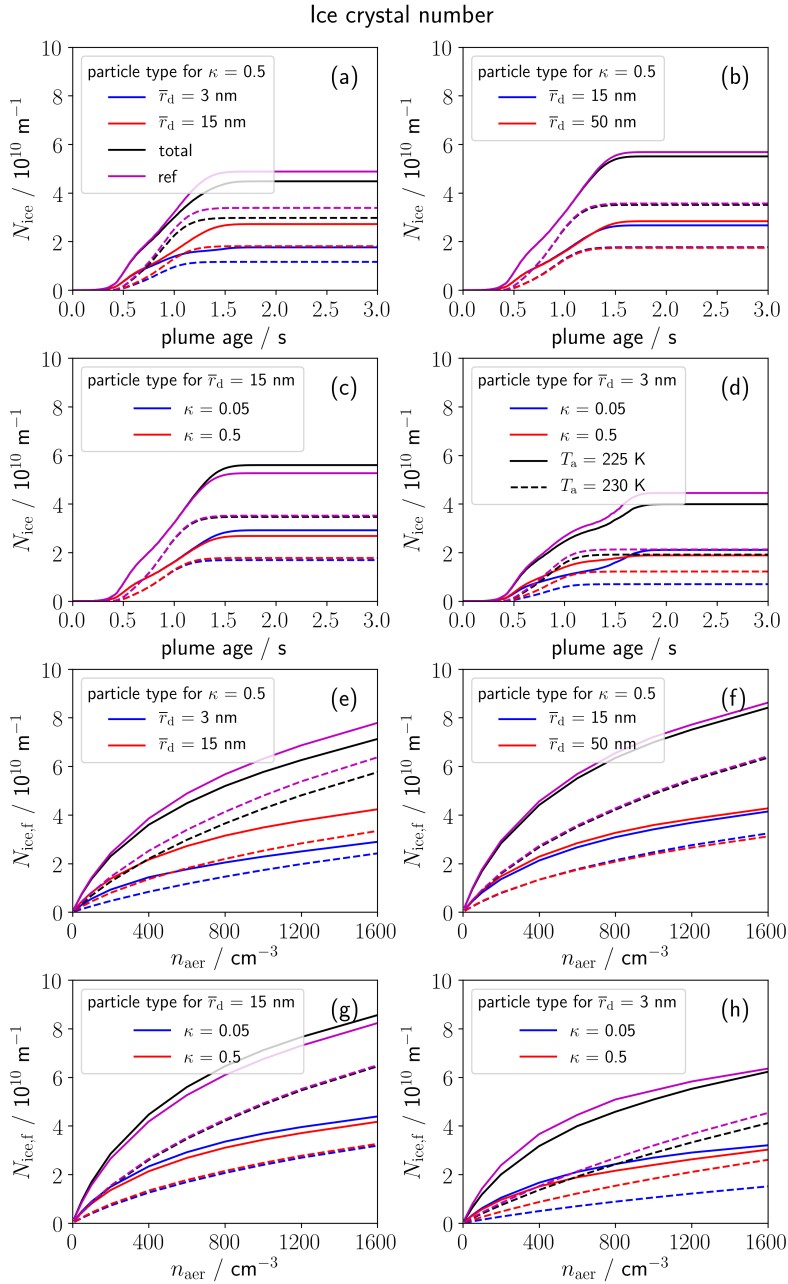

**Figure 6.** Ice crystal number concentration for two co-existing ambient aerosol particle ensembles: The first two rows show the temporal evolution of ice crystal number $N_{ice}$ and the last two rows the final ice crystal number $N_{ice,f}$ (at a plume age of 3 s) versus the total ambient aerosol number concentration of both ensembles. The blue and red lines depict $N_{ice}$ of the single aerosol ensembles and the black lines the sum of the both co-existing ensembles. The magenta lines represent reference cases from simulated single aerosol particle ensembles prescribing average quantities of the two co-existing ensembles. Panels (a)/(e) and (b)/(f) show results for a bi-modal aerosol size distribution (with mean dry radii $\overline{r}_d$ as displayed in the legends) for a fixed hygroscopicity parameter $\kappa = 0.5$. The other panels show results for two aerosol particle ensembles with a different solubility ($\kappa = 0.05$ and $0.5$) but a fixed $\overline{r}_d$, namely 15 nm in panels (c)/(g) and 3 nm in panels (d)/(h). All results are shown for two ambient temperatures (225 K solid and 230 K dashed.)

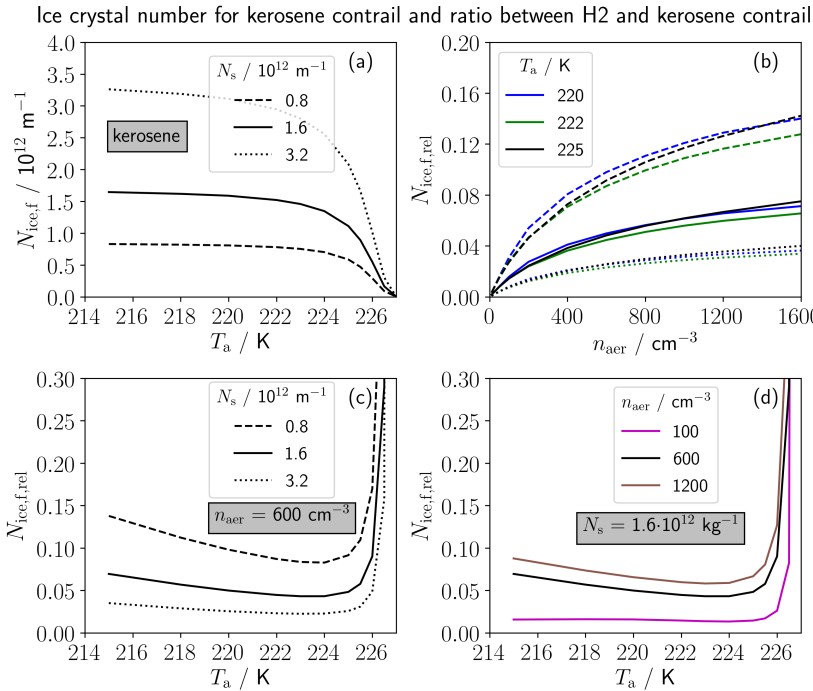

**Figure 7.** (a) Final ice crystal number of a conventional (kerosene) contrail versus ambient temperature ($T_a$) for 3 soot particle emission numbers per flight distance $N_s$ (different line styles). The other panels show the final ice crystal number of $H_2$ contrails relative to the final ice crystal number of kerosene contrails at same atmospheric conditions. Panel (b) shows $N_{ice,f,rel}$ versus ambient aerosol number concentration ($n_{aer}$) for the different soot number emissions and three $T_a$ cases (see legend (b)). Panel (c) shows the temperature variation of $N_{ice,f,rel}$ of one $H_2$ contrail (with $n_{aer} = 600\,\mathrm{cm}^{-3}$) with respect to three kerosene contrails with different $N_s$ as in panel (b). Panel (d) shows the temperature variation of three $H_2$ contrails (with different $n_{aer}$ as displayed in the legend) with respect to one kerosene contrail ($N_s = 1.6 \cdot 10^{12}\,\mathrm{m}^{-1}$). Thereby, the black solid lines in panels (c) and (d) represent the same case. The different line styles always refer to the three soot emission cases. The displayed quantities are given at a plume age of 3 s.





**Figure 8.** Contour plots (first two rows) of young contrail optical thickness $\tau$ over the trajectories' x-coordinate indicating the contrail width and the plume age. The first row show results for kerosene (with $N_s = 1.6 \cdot 10^{12}\,\mathrm{m}^{-1}$) and the second for the baseline $H_2$ case, both for ambient temperature ($T_a$) of 220 K on the lefthand and of 225 K on the righthand side. (e) shows the 90 Percentile optical thickness over the width $\tau_{90}$ for kerosene (red) and for $H_2$ prescribing three different ambient aerosol number concentrations (other colors in the legend) for $T_a$ = 220 K (dashed lines) and $T_a$ = 225 K (solid lines). The temporal evolution of $\tau_{90}$ has been smoothed using a running average method. For this analysis, we use the full FLUDILES trajectory ensemble of 25000 instead of the reduced ensemble of 1000.





## 5  Discussions

### 5.1  Plume contamination due to ultrafine oil particles

We expect that engine lubrication systems will be still used for $H_2$ engines causing emissions of oil vapors. Even though an air-oil separator recovers around 99% of the oil emissions, the residual may contaminate the engine plume. Ungeheuer et al.
(2022) have shown in laboratory experiments that "jet oil vapors reach gas-phase supersaturation in cooling emission plumes leading to rapid nucleation and formation of ultrafine volatile particles (UFPs) in the range of $\sim 10$–$20\,\mathrm{nm}$." These diameter ranges appear to be consistent with the ground-measured ambient UFPs downwind of the Frankfurt Airport, in which organic engine oil constituents have been identified (Ungeheuer et al., 2021).

The formed UFPs can contribute to droplet and ice crystal formation in addition to the background aerosol. Assuming a typical
oil consumption of around $1\,\mathrm{l\,h^{-1}}$ with 1% volume fraction (residual) that enters the plume, our estimates supply that the UFP number per flight distance could be even larger than that of soot particles. Since plume water-supersaturations are much higher for $H_2$ than for kerosene at same ambient conditions, we expect that droplets and ice crystals would mainly form on those UFPs rather than on the entrained ambient particles. This means that the number of ice crystals could be similar or even increased compared to conventional contrails. One should still keep in mind that the experiments of Ungeheuer et al. (2022) refer to
conventional kerosene combustion and the properties of the oil particles (size and chemical composition) could significantly change for $H_2$. Moreover, associate measurements at cruise altitude conditions are necessary which confirm the occurrence of those UFPs.

Finally, a hermetic and clean sealing of the engines from the oil system should be improved with regard to a complete jet oil recovery to achieve a valuable mitigation effort for contrail formation. The presented results in our manuscript are valid for
such an ideal sealing, where the formation of UFPs is excluded.

### 5.2  Scaling relations for different plume area evolution and fuel consumption

In the present study, we use the FLUDILES trajectory data (Vancassel et al., 2014) that were modified according to the description in Sect. 3.2. The presented results represent a single engine plume of an A340 aircraft. Extensive contrail quantities (like ice crystal number and mass) for the whole aircraft may be scaled with the number of engines (in our case 4). This scaling
is valid as long as we assume that there is no interference of the two exhaust plumes (on one side of the aircraft) during the contrail formation stage. This scaling could also be interpreted as a scaling with the ratio of the total fuel consumption ($m_{\mathrm{F,tot}}$) and our reference $m_{\mathrm{F}} = 0.58\,\mathrm{g\,m^{-1}}$ (for the baseline $T_\mathrm{a}$ and $p_\mathrm{a}$-values). Basically, one could plug in any reasonable value for $m_{\mathrm{F,tot}}$. However, this flexible fuel consumption scaling approach is only valid with several underlying assumptions that are usually not fulfilled. Firstly, one had to assume that the initial plume area $A_\mathrm{E}$ scales with the fuel consumption and, secondly,
that the plume dilution $\mathcal{C}(t)$ is independent of $A_\mathrm{E}$ and hence $A(t) \sim A_\mathrm{E}$.

However, Lewellen (2020) showed that the evolution of $\mathcal{C}$ itself depends on engine size. A different dilution does not only impact the plume area evolution but also the thermodynamic plume and the microphysical contrail properties (Lewellen, 2020; Bier et al., 2022). Neglecting those constraints and assuming fixed atmospheric conditions and aerosol particle properties,





we observe that $N_{\text{ice}} \sim N_{\text{aer}} \sim A(t) \sim A_{\text{E}} \sim m_{\text{F,tot}}$. This behavior is similar to kerosene contrails, where we find $N_{\text{ice}} \sim N_{\text{s}} \sim$
$m_{\text{F,tot}}$. Again, this is only a rough estimate with several underlying assumptions and for a fixed soot number emission index of
$EI_{\text{s}} = N_{\text{s}}/m_{\text{F,tot}}$.

So far, we have stressed that the given reference value of $m_{\text{F}}$ holds (only) for the baseline values of ambient temperature and
pressure. In our study, we use a constant $A_{\text{E}}$, which refers to a particular aircraft type with an engine nozzle radius of $0.5\,\text{m}$
(see Tab. 2). Moreover, we keep the plume exit temperature $T_{\text{E}}$ fixed in our setup. We repeat Eqs. (13) and (15) from Bier et al.
(2022), which follow from energy and mass conservation (Schumann, 1996; Schumann et al., 1998).

$$\mathcal{C}_{\text{E}} = \frac{Q\,(1-\eta)}{\bar{c}_{\text{p}}\,(T_{\text{E}} - T_{\text{a}})}, \tag{9}$$

$$m_{\text{F}} = \frac{A_{\text{E}}\,\rho_{\text{E}}}{\mathcal{C}_{\text{E}}} = \frac{A_{\text{E}}\,p_{\text{a}}}{\mathcal{C}_{\text{E}}\,R_{\text{d}}\,T_{\text{E}}}, \tag{10}$$

with $\rho_{\text{E}}$ denoting the plume air density at the engine exit, $R_{\text{d}}$ the specific gas constant for dry air and $\bar{c}_p = 1020\,\text{J(kg\,K)}^{-1}$ an
average specific heat capacity for dry air over a temperature range between around 200 and 600 K. From these equations it
follows that the initial dilution and, correspondingly, the fuel consumption implicitly change when $T_{\text{a}}$ takes a different value
($m_{\text{F}} \sim (T_{\text{E}} - T_{\text{a}})$). However, this $T_{\text{a}}$-dependence of $C_{\text{E}}$ and $m_{\text{F}}$ is much less crucial for $N_{\text{ice}}$ than the impact of $T_{\text{a}}$ on the plume
relative humidity evolution. A change in ambient pressure $p_{\text{a}}$ does not affect $\mathcal{C}_{\text{E}}$, but $m_{\text{F}}$ changes linearly with $p_{\text{a}}$. Hence, the
implied $m_{\text{F}}$-values in our $p_{\text{a}}$-sensitivity study differ non-negligibly. Clearly, these changes of the $m_{\text{F}}$-values are implications of
our choices in study design. In reality, the fuel consumption depends on the thrust setting and may change differently to how
we prescribed it, when $p_{\text{a}}$ and/or $T_{\text{a}}$ change.

### 5.3   Deliquescence relative humidity

We prescribe a fixed deliquescence relative humidity of the aerosol particles $DRH = 0.99$ in our set-up, a value close to water
saturation. The definition of a lower baseline value (according to Peng et al. (2022)) with appropriate sensitivity variations
would have been more reasonable. However, we achieved in several test simulations not robust results for $DRH < \approx 0.95$, in
particular for low mean aerosol particle dry size and hygrocsopicity parameter. This is likely due to one technical reason: While
water saturation is reached first near the plume edge and later on in the plume center in the first tenths of s, higher relative
humidities last longer in the plume center than at the edge towards the end of contrail formation (see e. g. Fig. 1 (d) of Bier et al.
(2022)). The lower the $DRH$-value, the longer is the potential time period for droplet and subsequent ice crystal formation, in
particular for lower ambient temperatures. The ambient particles are mainly entrained near the plume edge. Since we cannot
resolve this heterogeneous entrainment but ambient particles are mixed in for each trajectory with equal share, droplet and ice
crystal formation is likely overestimated and this overestimation increases with decreasing $DRH$.





## 6  Conclusions and outlook

In the recent past, several model studies investigated contrail formation behind commercial aircraft by means of analytical
approaches (Kärcher et al., 2015; Bier and Burkhardt, 2019), 0D box models (e.g., Kärcher and Yu, 2009; Vancassel et al.,
2014; Bier et al., 2022) and LES (e.g., Paoli et al., 2013; Khou et al., 2015; Lewellen, 2020). These studies focused on contrail
formation on soot particles on which the majority of ice crystals form for conventional engines (e.g., Kärcher and Yu, 2009;
Kleine et al., 2018). Switching to liquid hydrogen ($H_2$) propulsion, ice crystals are expected to form solely on background
particles mixed into the plume. Even though some of those studies account for ice crystal formation on background particles
(e.g., Kärcher and Yu, 2009; Kärcher et al., 2015; Lewellen, 2020), the implementation of this process is quite simplified
(e. g. activation relaxation approach in Kärcher et al. (2015)) and ambient particle properties are mostly kept fixed.

While Bier et al. (2022) have extended the particle-based Lagrangian Cloud Module (LCM; (Sölch and Kärcher, 2010)) by
contrail formation microphysics on soot particles, we here advance the LCM by specific contrail formation microphysics on
entrained background aerosol particles. The most relevant feature is that ambient particles are continuously entrained into the
plume instead of releasing a fixed number of soot particles. Moreover, we define an alternative droplet activation criterion and
improve the homogeneous freezing parameterization by accounting for the impact of the solution effect on droplet freezing.

Given the same atmospheric conditions and propulsion efficiency, the Schmidt-Appleman (SA)-threshold temperature (Schumann, 1996) is by around 10 K higher for $H_2$ than for kerosene fueled aircraft due to around 2.6 times higher water vapor
emissions for the same amount of released combustion heat. The homogeneous freezing temperature of water droplets is in
general smaller than the SA-threshold temperature for $H_2$ contrails and, therefore, becomes are more limiting criterion for
contrail formation as already pointed out by Gierens (2021).

The ice crystal number is strongly cut down for temperatures above around 230 K since smaller droplets do not freeze to ice
crystals any longer. Contrails cannot form anymore at temperatures above around 233–234 K in our study. While for kerosene
combustion the number of formed ice crystals approaches the emitted soot particle number for a sufficiently low ambient tem-
perature (e.g., Kärcher et al., 2015), the ice crystal number of $H_2$ contrails increases further with decreasing temperature. The
latter is because the water-supersaturation in the plume lasts longer for colder conditions and, hence, more of the entrained
aerosol particles can form droplets and ice crystals.

Our results highlight a large variability in the number of formed contrail ice crystals with varying ambient aerosol properties.
For a fixed particle size distribution and chemical composition, the ice crystal number clearly rises with increasing aerosol
number concentration. This increase becomes weaker for higher number concentrations ($> \approx 200\,\mathrm{cm}^{-3}$), in particular in a
colder environment. The variation of contrail ice nucleation with aerosol mean dry size and water solubility is low for larger
aerosol particles and high for small (mean radius $< \approx 10\,\mathrm{nm}$) particles. For these smaller particles, the sensitivity of the ice
crystal number with the water solubility of the aerosol particles is quite complex for lower temperatures due to various coun-
teracting microphysical processes.

In the real atmosphere, the background aerosol typically consists of multiple particle types with different mean dry sizes
(modes) and chemical composition. Therefore, we analyze contrail formation prescribing two co-existing aerosol particle en-





sembles that either differ in the mean dry size or hygroscopicity parameter. If these co-existing particle ensembles contain only larger (mean dry radii more than around $10\,\mathrm{nm}$) and well soluble aerosol particles, the ice crystal number for each of the ensemble can be estimated well from a simulation with appropriate single particle ensembles. The total ice crystal number of

the co-existing particle ensembles can be also approximated from one single particle ensemble prescribing average properties of mean dry size and solubility and the total number concentration of the co-existing particle ensembles. This is because the ice crystal number is not so sensitive to changes in this large particle range as mentioned above. Conversely, such an approach is not meaningful if a substantial fraction of small and weakly soluble aerosol particles (in particular nucleation mode particles) is present. If these particles co-exist with larger and/or better soluble particles, droplet and ice crystal formation on these particles

might be significantly supressed due to the competition effects between the aerosol particles. Due to the non-linearity between ice crystal number and mean dry size for small particles, the total ice crystal number of the co-existing particle ensembles might be significantly different to the average single particle ensemble.

Finally, we compare ice crystal formation, as a first measure of the mitigation potential, and visibility of $H_2$ contrails with conventional contrails. Varying both aerosol number concentration and soot number emissions, the $H_2$ contrail ice crystal num-

ber is significantly reduced (by more than 80–90%) compared to conventional contrails implying a great mitigation potential. This is mainly because ambient aerosol number concentrations are at least 1-2 orders of magnitude lower than soot particle number concentrations in young exhaust plumes behind conventional aircraft. For ambient temperatures only slightly below ($< \approx 0.5\,\mathrm{K}$) the SA-threshold temperature for kerosene combustion, the $H_2$ contrail ice crystal number can be higher since ice crystal formation on the weakly soluble soot particles becomes strongly limited by very low plume water-supersaturations

(e.g., Kärcher et al., 2015). The optical thickness is significantly decreased and the $H_2$ contrails either become slightly later visible or they might not be visible at all for low ambient aerosol number concentrations. On the other hand, $H_2$ contrails can form at lower flight altitudes (connected with ambient temperatures lying between the SA-threshold temperature for kerosene and the homogeneous freezing temperature of the water droplets) than conventional contrails, as also mentioned by Ström and Gierens (2002). In case of persistent contrails, this would increase the contrail coverage and partially compensate the mitigation

potential of hydrogen for the aviation climate impact.

*Data availability.* The presented data are available from the corresponding author upon request (Andreas.Bier@dlr.de).





## Appendix A: Numerical convergence and SIP merging

Once $RH_{\text{wat}}$ surpasses $DRH$ for the first time, a new SIP ensemble that represents the newly entrained dry aerosol is created
in every time step. In order to keep the total SIP number in an acceptable range, we employ a SIP merging technique where several similar-sized SIPs of the same category (aerosol, droplets and ice crystals) are merged into a single SIP. The merge operation is implemented such that the number and mass of the represented physical particles are conserved (Unterstrasser and Sölch, 2014). The SIP merging is executed when $N_{\text{SIP}}$ exceeds a certain threshold value (fixed to 1600 in our study). Then the new $N_{\text{SIP}}$ value is well below that threshold and starts to increase again. In the end, the SIP number follows a jigsaw pattern
as exemplarily shown in Fig. A1. The merge operation has some more (internal) parameters, e.g. how many SIPs are at most merged and what is maximum relative difference between SIPs that are merged. We experimented with those parameters and found numerical convergence, that means our present configuration yields basically identical results with simulations with higher $N_{\text{SIP}}$.

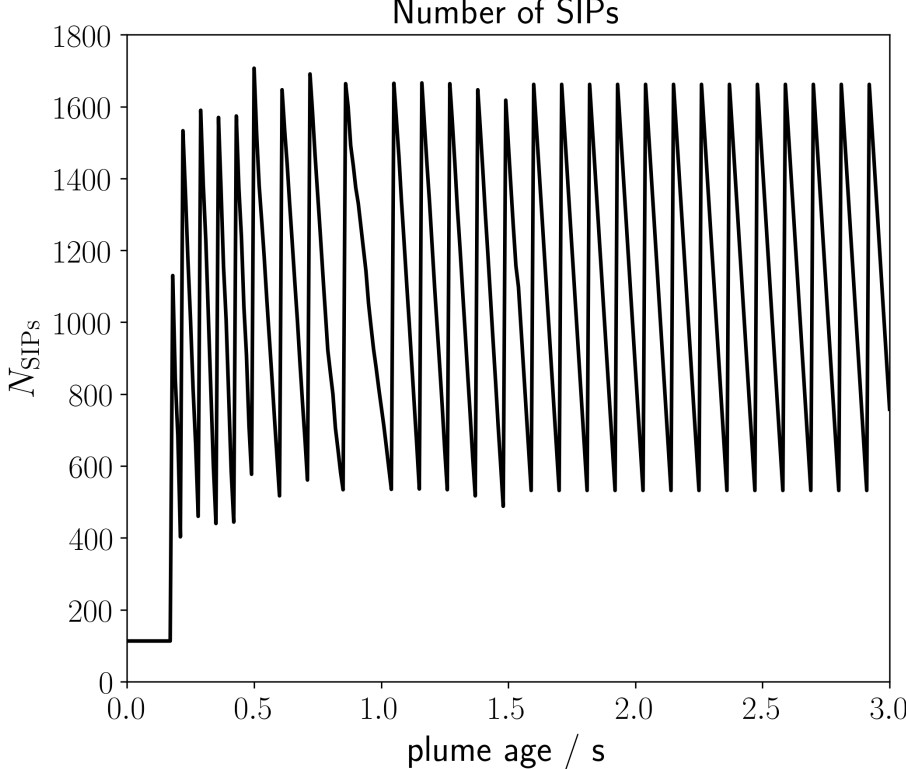

**Figure A1.** Temporal evolution of the number of Simulation Particles ($N_{\text{SIPs}}$) for a contrail forming on ambient particles using one average FLUDILES trajectory at baseline conditions (see Tab. 2).



**Appendix B: Parameterization of homogeneous freezing temperature**

We calculate the homogeneous freezing temperature as

$$T_{\mathrm{frz}} = T_{\mathrm{frz},0}(V_{\mathrm{wat}}, \dot{T}) - \Delta T(a_{\mathrm{wat}}), \tag{B1}$$

where $T_{\mathrm{frz},0}$ is the freezing temperature assuming a pure super-cooled water droplet ($a_{\mathrm{wat}} = 1$). It is determined from Eq. (6) of Bier et al. (2022) following the approach of Kärcher et al. (2015) and Riechers et al. (2013) and varies with droplet water volume $V_{\mathrm{wat}}$ and cooling rate $\dot{T}$.

The second term is a correction term based on the study from O and Wood (2016) approximating the decrease in $T_{\mathrm{frz}}$ with decreasing activity of water $a_{\mathrm{wat}}$ (e.g., Koop et al., 2000). O and Wood (2016) developed an approximation for the homogeneous freezing temperature of solution droplets ($T_{\mathrm{frz,owood}}$) as a function of water volume and activity, however neglecting the cooling rate. Thereby, that temperature is iteratively calculated for which the mean number of critical embryos becomes equal to one triggering the freezing process in the droplet (see their Eq. (1)). For our activated water droplets ($a_{\mathrm{wat}} > 0.90$), we find that the

parameterization yield robust results for droplet radii $r >= 1\,\mu$m. On the other hand, the solution effect is frequently important for smaller droplets ($r < 1\,\mu$m) formed on freshly entrained aerosol particles.

For simplicity, we prescribe a fixed droplet radius r = 1 $\mu$m for the estimation of our correction term. We evaluate $T_{\mathrm{frz,owood}}$ for different $a_{\mathrm{wat}}$ values that result from a variation of $r_d$. Setting $\Delta T = T_{\mathrm{frz,owood}}(a_{\mathrm{wat}} = 1) - T_{\mathrm{frz,owood}}(a_{\mathrm{wat}})$ and $\Delta a_{\mathrm{wat}} = 1 - a_{\mathrm{wat}}$, we define the following quadratic fit function

$$\Delta T = a \cdot \Delta a_{\mathrm{wat}}^2 + b \cdot \Delta a_{\mathrm{wat}} + c, \tag{B2}$$

which is valid for $a_{\mathrm{wat}} >= 0.90$ with the fit parameters $a = 345.746$, $b = 100.977$ and $c = 0.01687$ and a square root mean error $R^2 = 0.99999$. Even though the expression for $\Delta T$ is a simplified correction term, Eq. (B1) combines the sensitivity of the homogeneous freezing to all major effects.

Fig. B1 shows the variation of $T_{\mathrm{frz}}$ with droplet radius $r$ and water activity for a plume cooling rate of -10 Ks$^{-1}$. For $a_{\mathrm{wat}} = 1$

(representing pure water droplets), $T_{\mathrm{frz}}$ equals to $T_{\mathrm{frz},0}$ and ranges between around 229 and 233 K. Thereby, $T_{\mathrm{frz}}$ increases with rising $r$. The strong decrease of $T_{\mathrm{frz}}$ with decreasing $a_{\mathrm{wat}}$ (for fixed $r$) down to around 215 K emphasizes the importance of accounting for the solution effect in droplets. In contrast, the impact of a varying cooling rate is low (not shown).




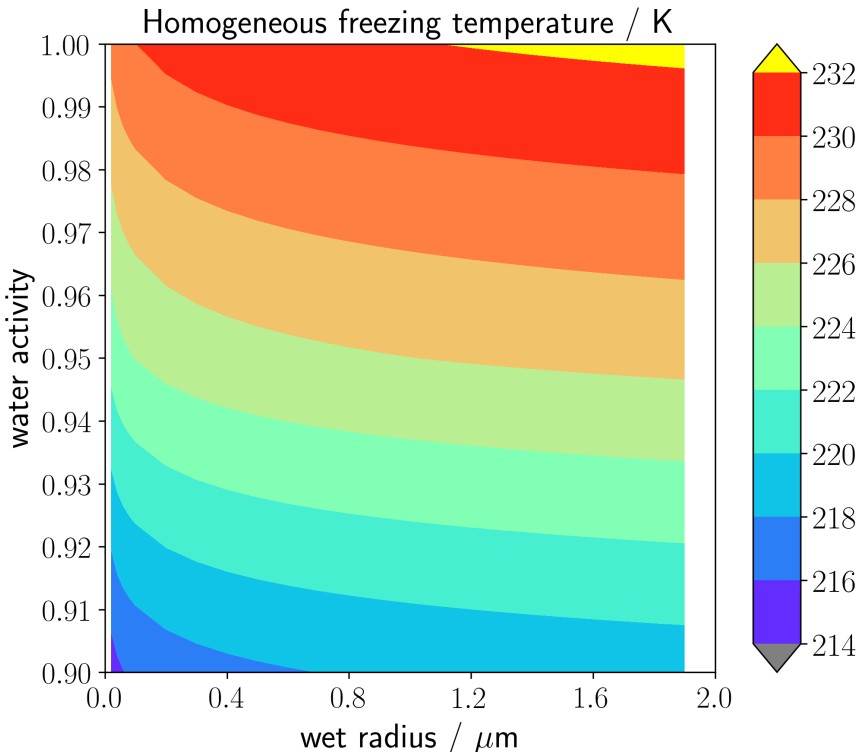

**Figure B1.** Contour plot showing the homogeneous freezing temperature of supercooled solution droplets over the wet radius and water activity. The cooling rate is set to -10 Ks$^{-1}$.



*Author contributions.* AB performed the simulations, created the tables and figures and wrote the first draft of the manuscript. AB and SU conceptualized the study; they evaluated and interpreted the results. AB, SU and TJ wrote and edited the manuscript. AB, SU, JZ, DH and 730 AL advanced and extended the box model code.

*Competing interests.* The contact author declares that there are no competing interests.

*Acknowledgements.* The scientific work has been funded by the Deutsche Forschungsgemeinschaft (DFG) within the project "BI 2128/1-1" and by the DLR project "H2CONTRAIL". We thank X. Vancassel for providing the original FLUDILES trajectory data set.



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
