# Peer review of "Contrail formation on ambient aerosol particles for aircraft with hydrogen combustion: A box model trajectory study"

_EGUsphere, 2023_

## Author Comment (AC1)

This document contains the reply to the Editor and to both reviewers.

**Reply to the Editor**

**Dear Editor,**

Many thanks for granting us more time for revising the manuscript. Furthermore, we appreciate the constructive and helpful comments of both reviewers.

In the first section of this document we report changes to the manuscript irrespective of the reviewer comments. Afterwards, the point.to-point replies to both reviewer comments and the revised manuscript with track changes follow.

During the revision, we came across a flaw we made in computing the fuel consumption and the initial plume mass flow.

- In the end, the ice crystal number per flight distance as shown in most of our figures had to be scaled up by a constant factor of 1.92. Hence, all figures were updated. The description had to be changed only on few occasions, where we mentioned absolute numbers. As all plotted curves are scaled by the same factor, any statement on relative differences between various curves remains true and there was no need to change them. Furthermore, the kerosene cases have to scaled in the same way. Hence, the comparison between H2 and kerosene contrails give the same results.
- We introduced the new subsection 3.3.2 "**Plume cross-sectional area**" describing in some detail our new findings. Looking at previous papers (also of other groups) we realized that the definition of the plume area is a bit intricate. Hence, we aimed at defining everything properly and write out all implicit assumptions. We also moved some paragraphs from the sections 3.3.2 and 5.2 to this new subsection.

**RC1: First response to reviewer 1**

Dear reviewer,

thank you very much for your positive evaluation and the important comments and suggestions. Please find our answers below (the reviewer's comments are repeated in italics).

This is an excellent and timely analysis of the expected contrail behavior from an aviation gas turbine engine burning H2 fuel. There is increasing interest in this topic as H2 is being considered as a potential, longer-term solution to the problem of burning of fossil fuels by aviation. There are many questions and issues to adopting H2 fuels, but the changes in contrail properties are correctly identified by the authors as a key one, since contrails are such an important part of aviation's radiative impact.

I think the paper is very well written and the study very well executed and the paper can be published after addressing several small issues.

1. In section 2.1, the authors note that chemical ion nucleation is important source of particles in the UT. Some have, in the past, considered ion-nucleation in the aircraft exhaust plume. Since there are no soot particles and the current model is assuming nucleation on

ambient particles, it is probably worth mentioning that ion nucleation from combustion chemi-ions is not being considered in this analysis. I don't think that is a shortcoming of the analysis, but it should be made clear that that may be another nucleation pathway, but is not being considered here (perhaps add in section 3.3?). If the authors have explicit reasons why they chose not to do so, that may be useful to add, but not necessary.

This is an important aspect. In fact, one important formation mechanism for ultrafine volatile particles in aircraft plumes is the recombination of chemi-ions to electrically charged or neutral molecular clusters.

We decided to generalize the headline of Sect. 5.1 to "Potential sources for the formation of ultrafine volatile particles" and subdivided this in Sect. 5.1.1. "Nitrogen compounds" and the already existing Sect. 5.1.2 "Ultrafine oil particles". We have included following text in Sect. 5.1.1:

"Several recent studies considered chemi-ions that are mainly composed of sulfur species (e.g., Yu and Turco, 1997). While sulfur is likely not produced during H2 combustion, NOx is still emitted. The reaction of NOx with H2O and OH-species potentially leads to the formation of nitrogen compounds like nitric acid that were observed in conventional aircraft plumes (e.g., Tremmel et al., 1998). Moreover, Wang et al. (2020) have shown within cloud chamber experiments that nitric acid and ammonia can nucleate directly to form volatile ammonium nitrate particles at temperatures below 258 K.

Finally, nitrogen species might be a potential source for the nucleation of ultrafine volatile particle both in conventional and H2 combustion plumes, but the formation process behind is not yet sufficiently understood."

2. In section 4.2.2 line 433-434, the authors note "This behavior is similar to . . . kerosene combustion." I think it is notable and a bit remarkable that this is the case, despite the very different nuclei number density behavior (kerosene starts high and dilutes, while the H2 case has continual entrainment). I think a phrase should be added to emphasize that the dependence is similar DESPITE quite different nuclei number histories.

This is an important point. We have added "...despite the very different temporal evolution in the exhaust particle number concentration" after "similar to kerosene combustion".

3. Figure 6 is an important result. However, I found the caption confusing. The legends do not have the T = 230 K line type (dashed) identified. And the caption only mentions this at the very end. I was trying to understand the first two panels and was reading that part of the caption and not finding out to what the dashed lines correspond. I would recommend putting the statement about the two temperatures at the beginning of the caption to avoid this problem, if they don't want to have the dashed lines in the figure legend itself.

We apologize for the confusion. We have moved the caption description "All results are shown for two ambient temperatures (225 K solid and 230 K dashed)" at the very beginning and we have created a separate temperature legend at the top of the figure.

4. Finally, section 5.1 is a bit inappropriate, I think. First, the analysis doesn't really do anything with oil particles. So this is sort of a side comment that this might need to be studies more. And the paper only cites one paper on oil, while there is significant literature on aviation engine oil contributions to aviation PM emissions. And some of the other literature presents data significantly distinct from the one paper cited. The authors should either review the literature more broadly or minimize their discussion of the oil emission data. And the comment that "a hermetic and clean sealing of the engines from the oil system should be improved with regard to a complete jet oil recovery" indicates a lack of understanding of how

the oil system on an aviation gas turbine engine functions. A comment along the lines of "reduction or elimination of oil emissions from the aircraft engines may be . . . a valuable mitigation effort for contrail formation" might be better. It should also be qualified ("may") since oil's role in contrails is still very poorly understood.

We now also mention a study by Yu et al, 2010 in the Introduction. We had discussions with aircraft and engine manufacturers and they confirmed that the amount of emitted oil could be reduced by technical improvements, but "complete recovery" may sound too strong as there will likely always be losses to the environment. We changed the sentence to "Finally, a hermetic and clean sealing of the engines from the oil system aiming at a complete jet oil recovery could be a technical means to achieve a valuable mitigation effort for contrail formation if future model studies and flight campaigns give hints at abundant droplet formation on oil particles."

But clearly the role of oil droplets is poorly understood. And we hopefully convey this message sufficiently clearly in the manuscript.

The second reviewer had a quite different comment on the section about oil droplets and wanted that section to be expanded. See our reply to the other reviewer as well.

**RC2: Second response to reviewer 1**

One more very minor comment from Reviewer 1:

line 454 "can easier form water droplets." might read better as "can form water droplets more easily."

Thank you for the minor comment. We have changed the sentence accordingly.

**RC 3: Response to reviewer 2**

The manuscript explores how contrails might form and persist for zero-particle emissions engines with enhanced water vapor emissions, which are conditions relevant for hydrogen propulsion. Particle size distributions and hygroscopicities are assumed based on recent literature in order to parametrically understand the impacts on contrail ice crystal number throughout the near-field plume evolution. Interestingly, the results demonstrate that the increase in water vapor emissions and relatively low concentrations of available cloud condensation nuclei (CCN) allow contrails to form at warmer temperatures than for conventional kerosene burning engines, which could mean that a transition toward hydrogen propulsion results in an increase in climate-altering contrails. Simulations are carried out with both one and two lognormal size distribution modes, and the results are interesting for understanding the relative role of new particle formation versus Aitken and accumulation mode particles. These results hint that new particle formation from engine oil vapors may play a role in forming contrails; although, the choice of simulation parameters (e.g., forcing the concentrations of both modes to be the same) make it hard to understand how these particles may influence a contrail behind a hydrogen engine. Overall, the manuscript is well written and is an interesting and useful contribution to ACP and the contrail literature. I'd be inclined to recommend the paper for publication after the following comments are satisfactorily addressed:

1) The authors demonstrate awareness of recent work hinting that the venting of oil vapors may be a significant source of CCN for contrail formation in the "soot-poor" emissions regime, and the discussion in Section 5.1 is a good and important addition to the paper. However, I think not applying the model to directly explore the potential influence of these particles is a major gap in the present study. Could the 2-mode simulation be set up to create an additional figure to better understand how the contrail ice crystal number would change if a constant particle emissions source size mode from oil vapor with the following parameters were externally mixed with the ambient baseline particle number size distribution mode from Brock et al., 2021 and Section 4.3.1:

- Mode 1 (New Particle Formation From Continuously Vented Oil Vapor):
  - $r_d(nm) = 5$ , sigma = 1.2
  - $\circ$  kappa = 0
  - Emissions Index (number of particles emitted from 2 engines per second) =  $10^{11}$ ,  $10^{13}$ ,  $10^{15}$
- Mode 2 (Baseline ambient lognormal particle mode from Table 2):
  - $r_d(nm) = 15 nm, sigma = 1.6$
  - $\circ$  kappa = 0.5
  - $\circ$  Concentration = 600 cm-3

The fundamental question is, do large numbers of small wetable, but insoluble particles dominate the contrail ice number in the presence of relatively few, large, sulfate particles from the ambient atmosophere?

We wanted to follow your recommendation and include simulations with oils droplets. After intensive discussions within the author team and other persons, we concluded that it is better to defer the "oil topic" to a future study. Several reasons led to our decision:

- 1. The inclusion of oil droplets in the box model is not as simple as it may appear. It is not possible to simply set kappa of the oil droplets to 0. In this case, the Köhler curve has no peak as the solution term drops out (hence no critical radius can be determined). Moreover, the water activity would be 1. According to our definition of droplet activation (awat>awat,c), such droplets would be activated from the beginning on, which also makes no sense. Hence, it scientifically sounder to implement the adsorption approach in our model (see also our comment to your question 10). The competition effects between emitted oil droplets and entrained ambient aerosol particles may be intricate and can depend sensitively on choices in the microphysical modelling of each particle type. Hence, this would need much extra text to describe it in sufficient detail. In our impression the manuscript in the present version is already quite long and we deem it out-of-scope to describe the implementation, the new results and their interpretation in sufficient detail.
- 2. There are different opinions about how much oil is lost to the environment and on the size of them. Moreover, measurement studies up-to-date can clearly measure only kerosene plumes, where vaporized and recombined oil droplets may interact with other plume species. In "purer" H2 plumes, oil droplets may in the end look different to those in kerosene plumes and have other properties affecting their activation propensity. Hence, box model simulations with oil droplets would be very speculative and a cautious interpretation is needed.
- 3. [Clearly, one could change the title of our manuscript, but:] With the title we wanted to stress from the beginning on, that (only) ambient aerosol is considered. We believe that our study is an important step towards understanding the impact of aerosol

properties and variability on H2 contrail formation (clearly with the assumption, that oil droplets are negligible; even if oil droplets have an effect, they may not be emitted continuously along the whole flight). Our study focuses on analyzing the impact of many particle properties and also the interaction of different aerosol particle types, which is novel. And we believe we should keep this clear focus on ambient aerosol.

4. If upcoming in-situ measurements of H2 plumes/contrails hint at an important role of oil droplets in contrail formation, it is clear that our next study in this research topic has to turn the spotlight on oil droplets. And the present manuscript will serve as a good starting point for this.

2) Line 2: Change "has" to "have"

Done.

3) I find the use of the word "ensemble" to describe distinct size distribution modes to be confusing. I'd suggest replacing the word "ensemble" with "lognormal size mode" on Line 19 and throughout the manuscript.

You are right that "(lognormal) size modes" would be the more appropriate phrase to describe distinct particle size distributions. The majority of our analysis is based on prescribing one single aerosol particle mode anyway. In Sect. 4.3.2., we also investigate contrail formation for ambient particles prescribing two coexisting aerosol particle *ensembles*.

Thereby, we either prescribe two different lognormal size modes with a fixed hygroscopicity or one size mode but with two different hygroscopicity values. In the latter case, it would be misleading to talk about "two distinct size modes". That is why we decided to use the more general word "ensemble", which enables the differentiation of aerosol particle types both in their size mode or their solubility. For introducing the terminology more clearly, we replaced the sentence at the beginning of Sect. 4.3.2 as follows:

"We restrict our analysis to a scenario, where the two co-existing particle ensembles always differ either in their mode (in our case  $r_d$ ) or the solubility kappa."

4) Lines 33-34: Suggest "in particular on soot particles relative to co-emitted organic-sulfate particles"

Thank you for the suggestion. We have changed the phrase accordingly.

5) Line 50: It would also be good to cite the field studies showing detection of oil signatures in engine particles for high-soot engines, albeit, with a focus on mass spectral detection methods that only see particles with diameters > 100 nm:

- 1. Yu et al., 2010: https://doi.org/10.1021/es102145z
- 2. Yu et al., 2012: https://doi.org/10.1021/es301692t

Thank you for this advice. We have included following text behind the Ungeheuer et al. (2022) citation:

"In ground field measurements, lubrication oil droplets with volumetric mean dry radii ranging between around 125 - 175 nm were observed by sampling directly from the breather vents (Yu et al., 2010). Moreover, Yu et al. (2012) performed the first field study that investigates in-flight lubrication oil emissions behind a commercial aircraft. Thereby, they find a significant contribution of lubrication oil constituents in organic particulate matter emissions from the engine exhausts that are typically associated with high soot number emissions."

6) *Line 74: Change "magnitudes" to "magnitude"* Done.

7) Line 78: large variability in atmospheric aerosol properties and co-emitted volatile particles from engine oil vapor We include the sentence:

"Moreover, it is not clear whether ultrafine volatile particles originating from lubrication oils and NOx emission play a role in droplet and ice crystal formation."

8) Line 130: It was not clear to me that the concentrations reported by Beer et al., 2020 in their supplement were for non-volatile particles. Concentrations of 200-300 cm-3 seem high to me for dust and black carbon number. Please double check this.

Yes, we agree with that. It was not clear whether the data show non-vol or total Condensation Nuclei (CN) as the reference to the previous plot was misleading. We received confirmation from C. Beer that these values refer to measurements of total CN number concentrations.

**We changed the text to**

"They show (in their Supplement) altitude profiles of number concentrations with average values between 200 and 300 cm-3 for total Condensation Nuclei (CN) with dry radii larger than 5 nm."

9) *Line 140: Suggest "higher" instead of "better"* Changed, thanks.

10) Section 3.3.1, how would treating uncoated soot or dust as in soluble but able to adsorb water change things versus standard Köhler Theory, if at all? See, e.g., Kumar et al., 2009 (https://acp.copernicus.org/articles/9/2517/2009/).

Thank you for this important hint. We have included following paragraph in Sect. 3.3.1 after the discussion of heterogeneous ice nucleation:

"Insoluble but still wettable particles like uncoated soot and oil droplets could be also treated by adsorption activation theory. It is based on the standard Köhler theory but uses a specific description of the water activity that accounts for adsorption processes. In particular, the Frankel, Halsey and Hill (FHH) adsorption approach (Sorjamaa and Laaksonen, 2007; Kumar et al., 2009) is able to treat multilayer adsorption of water vapor onto insoluble particles and should be considered in future studies."

11) Line 292: insert "is" to read "expression is the Kelvin term" Done.

**12) Line 325: Does this factor of 4 assume a 4-engine aircraft?**

Yes, it does. We have included the information in the sentence: "Since  $A_E$  is the area of one engine nozzle exit plane (based on the FLUDILES data for the 4-engine A340-300 aircraft and kept constant in this study), ...". Moreover, we made this point clearer in the figure caption by stating that we refer to the plume of a single engine and not the full contrail.

13) Line 364: is this because the soot particles are already well mixed with the plume water vapor?

Yes, they are. Soot particles are typically formed right behind the engine exit plane and, hence, at plume ages where plume relative humidity is still very low and no activation occurs. In contrast, ambient particles are continuously mixed in the plume and, therefore, the number of entrained ambient particles increases with plume age (see e.g. Fig. 2 (c)).

We have modified the sentence as follows:

"This is different to exhaust species like soot particles since they typically form right behind the engine exit (when plume relative humidity is still low and no activation occurs) and their emitted number (per flight distance) is then assumed to be constant over time."

14) Line 421: parameter values is hyphenated and misspelled Thanks. Done.

15) Lines 451-457: It's fascinating to think about this discussion in the context of low-kappa oil-nucleated particles that might be emitted by a hydrogen engine; although, it is not exactly the same set of parameters (as discussed above). I hope the authors will carry out the additional simulations to inform the interplay between new particle formation from co-emitted vapors and ambient aerosols.

The complexity of microphysical phenomena described in lines 451-457 is one reason why we would like to defer the oil topic to a future study. In our feeling, there would not be enough room for discussing the competition between droplet formation on oil droplets and ambient aerosol particles with the desired level of detail.

*16) Lines 544: Strike "by" + 17) Line 545: Replace "by" with "is"* Done.

18) Line 650: Strike "are"We have replaced "are" by "a" such that the phrase sounds

"... becomes a more limiting criterion for contrail formation..."

---

## Author Response (AR2)

**Dear Editor,**

**We very much appreciate your diligent reading and spotting of typos! Thank you very much also for your efficient handling of the manuscript over the whole review stage.**

**We followed all your recommendations. Only those items, where we thought an answer would be helpful for you, are answered below.**

**In excess of the changes you requested, we had to change Figure A1 very slightly, as we realised that the figure version in the revised manuscript had a clipped axis label.**

**Best wishes,**

**Simon Unterstrasser (on behalf of the author team)**

Abstract: Consider shortening the abstract. The length should be about 250 words (see ACP guidelines) and yours has currently 444 words.

> **We shortened it considerably from 444 to 357 words.**

P5, L135: References should be embedded in the text -> change parenthesis -> Beer et al. (2020)

P8, L213: remaineder -> remainder

P11, L292: You refer here to the previous version of your manuscript? In that case you should not write it like this in the current version of the manuscript. Mentioning this is only important in the reply to the referees, but not in the current version of your manuscript.

> **This made a reference to our previous paper, not the previous version of the current manuscript. It is already explicitly written in this sentence.**

P12, L338: Add here after equation 9 "for A(t)" or for the "effective area A(t)".

P13, L367: Parenthesis around the reference not correct. It should read "Kulmala et al. (1993).

P13, L366: Eq -> equation (here it should not be abbreviated).

P14, L387: I guess something went wrong here. Who is O? I guess the rest of the last name got somehow lost.

> **The last name has indeed only one character!**

P14, L393 and throughout the manuscript: Tab -> Table. Note usually Table is not abbreviated (see ACP guidelines).

P14, L398: Sec. -> Sect.

P15, L416: Why 3 and 5s? Can you give a motivation why this short time is sufficient?

> **We added an explanation.**

P20, L526 and throughout the remainder of the manuscript: Add always also which figure you are refereeing to. The way to mention the respective panels should be e.g Fig 6a, Fig6b and so on.

P21, L534: appendix -> Appendix

P25, L623: add used -> of the cryoplanes used in our …….

P25, l624: study -> studies (?)

P27, Figure 6 caption, line 4: remove "the" -> sum of both co-existing ensembles

P31, L685: What is your case 4? Please provide more details on the respective case or give the reference to the respective section.

P35, L779: What is the abbreviation "SIP" standing for?

> **The abbreviation SIP was introduced in section "3.1 LCM box model"**

P35, Figure A caption: Simulation Particles -> simulation particles.

> **We introduced the abbreviation SIP here again to help the reader.**

P36, L796 and 797: check reference.

> **"O and Wood" is correct.**

[revised manuscript text omitted]
_{\text{ice,f}}$ strongly increases with decreasing $T_{\text{a}}$ and then approaches the respective $N_{\text{s}}$-values for sufficiently low $T_{\text{a}}$. For higher $N_{\text{s}}$, the number of formed ice crystals rises more steeply and approaches $N_{\text{s}}$ at lower $T_{\text{a}}$.

Now, we investigate the ratio of the ice crystal numbers between H$_2$ and kerosene contrails ($N_{\text{ice,f,rel}}$) shown in panels (b)–(d). We constrain our analysis to that temperature range in which kerosene contrails are able to form according to the SA-criterion. A mitigation is achieved for $N_{\text{ice,f,rel}} < 1$, where a lower value is connected with a higher mitigation potential. Panel (b) shows

605 $N_{\text{ice,f,rel}}$ versus $n_{\text{aer}}$ for the three $N_{\text{s}}$ (see line style) and for three $T_{\text{a}}$ cases (different colors). In general, we see a clear decrease in $N_{\text{ice,f,rel}}$ with increasing $N_{\text{s}}$ and decreasing $n_{\text{aer}}$. Thereby, $N_{\text{ice,f,rel}}$ is below 0.1 for $n_{\text{
[revised manuscript text omitted]